# Geometric Uncertainty for Detecting and Correcting Hallucinations in LLMs

## Abstract

Large language models are known to hallucinate, generating linguistically plausible but incorrect answers to questions. Uncertainty quantification has been proposed as a strategy to detect such behaviour, but existing methods lack a unified framework to assess reliability at both the prompt and answer level. We introduce a geometric framework which quantifies language model uncertainty at both levels by explicitly modelling a prompt-conditioned semantic distribution in answer embedding space. Our approach is black-box and sampling-based; we generate multiple answers per prompt, and use archetypal analysis to estimate a geometric support for the answer distribution. At the prompt level, we approximate the distribution entropy to quantify uncertainty; for each individual answer, we then use notions of atypicality to assess its reliability relative to the batch. We employ our framework to not only detect hallucinations but correct them, by selecting the batch example deemed most reliable. Experiments show that our framework performs comparably to or better than prior methods on short form question-answering datasets, and achieves superior results on medical datasets where hallucinations carry particularly critical risks. Beyond pure performance, we suggest the theoretical grounding of our work provides support for semantic distributions as useful objects of study for language model uncertainty.

## 1 Introduction

Large language models (LLMs) have achieved remarkable performance across diverse natural language processing tasks (Guo et al., 2025; Anthropic, 2025; Gemini Team, Google DeepMind, 2025; OpenAI, 2025) and are increasingly applied in areas such as medical diagnosis, law, and financial advice (Yang et al., 2025; Chen et al., 2024; Kong et al., 2024). Hallucinations, however, where models generate plausible but false or fabricated content, pose significant risks for adoption in high-stakes applications (Farquhar et al., 2024).

Uncertainty quantification (UQ) methods have been proposed to detect when models are producing unreliable outputs (Liu et al., 2025; Xiong et al., 2024). Effective UQ can serve as a critical layer of security and transparency, enabling systems to flag problematic responses and helping users exercise caution when reliability is compromised (Farquhar et al., 2024). This capability is particularly crucial for applications such as healthcare and legal services where incorrect information can cause significant harm (Huang et al., 2025; Asgari et al., 2025; Latif, 2025).

UQ methods can generally be divided into two groups: sampling-based and activation-based. Sampling-based methods generate multiple responses for a prompt and estimate uncertainty over the resulting batch, often requiring only black-box access. Activation-based methods use internal model representations to estimate uncertainty at the single-response level, which requires white-box access. Previous work also distinguishes between external and internal uncertainty, the former arising from ambiguous queries and the latter from insufficient model knowledge (Li et al., 2025).

Within sampling-based methods, where a batch of responses is generated per prompt, we additionally distinguish between two types of uncertainty estimate. A *global* estimate summarizes the uncertainty of the batch, whereas a *local* estimate scores each individual response relative to the others in the same batch. Prior work has referred to the latter as 'confidence' (Lin et al., 2024). Global UQ can allow prompt-level

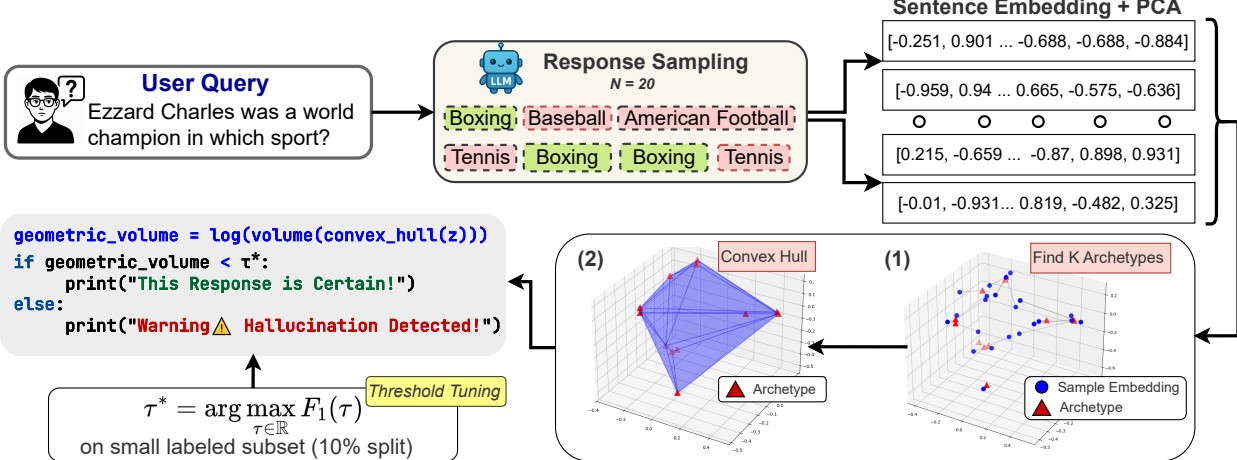

Figure 1: A schematic of geometric volume: (1) sample $n$ responses from the LLM, (2) embed and apply dimensionality reduction, (3) perform archetypal analysis and compute the convex hull, and (4) apply a threshold to detect hallucination.

risk mitigation such as abstention or triggered context retrieval (Wen et al., 2025). In high-stakes settings, local UQ enables the selection of the most reliable response from several alternatives, thereby reducing the risk of harmful hallucinations.

We introduce a geometric framework to quantify global and local uncertainty using only black-box access. Our approach is sampling-based: for a given prompt we generate a batch of responses at non-zero temperature and embed them with a sentence encoder. We then apply Archetypal Analysis (AA) (Cutler & Breiman, 1994) to identify a set of archetypes that span the embedding space, approximating a geometric support for the semantic distribution. Our first contribution is *Geometric Volume*, a global uncertainty metric defined as the convex hull volume of the archetypes, which reflects the semantic spread of the batch.

Our second contribution is *Geometric Suspicion*, a local uncertainty measure that quantifies response atypicality within the sampled distribution. Our measure is derived from three complementary views of atypicality, and offers a geometrically grounded alternative to heuristic graph measures for differentiating high and low uncertainty responses.

We validate our framework on `CLAMBER`, `TriviaQA`, `ScienceQA`, `MedicalQA`, and `K-QA`. We also provide theoretical analysis linking convex hull volume to entropy. We summarize our contributions as follows:

- We propose a unified framework to quantify global and local LLM uncertainty with only black-box access, using an estimated geometric support for prompt-conditioned semantic distributions.

- We introduce *Geometric Volume*, a global uncertainty metric which quantifies the semantic distribution entropy. We prove our metric yields an upper bound on the entropy of distributions defined within the estimated geometric support.

- We construct *Geometric Suspicion*, a local uncertainty metric integrated within the same geometric framework, which reduces hallucinations by guiding best-of-N response selection.

## 2 Related Work

**Semantic Volume:** Our work is closely related to Semantic Volume (Li et al., 2025), which also analyzes the semantic dispersion of a batch of natural language outputs. Specifically, Semantic Volume computes the determinant of the Gram matrix formed from a batch of embeddings, where a low value corresponds to low uncertainty and vice versa. This value measures the volume of the parallelepiped formed by the embedding

vectors. We prove in Section 3.3.2 that our archetype-based method leads to a tighter bound on semantic distribution entropy, whilst offering useful intermediate representations for calculating local uncertainty.

**Convex Hull Approaches:** Several recent works use convex hull area over response embeddings as a proxy for uncertainty, projecting into two dimensions and summing hulls around clusters (Catak & Kuzlu, 2024; Catak et al., 2024). These works, however, vastly simplify the semantic space by projecting into only two dimensions with principal component analysis. They also first cluster responses before computing and summing the convex hull area around each cluster, which ignores how far apart in embedding space separate clusters may be. In doing so crucial information is lost regarding the variation in response meaning a model is likely to produce given a particular prompt.

**Semantic Entropy and Self-Consistency:** Semantic entropy detects hallucinations by clustering semantically equivalent responses using bidirectional entailment, then computing entropy over those clusters (Farquhar et al., 2024). Self-consistency methods have built on their work and are emerging as a dominant paradigm (Taubenfeld et al., 2025; Wan et al., 2025; Savage et al., 2024), where multiple responses are sampled and their agreement is used as an uncertainty signal. These works address global UQ but typically do not provide additional methods for local UQ.

**Semantic Space Confidence:** Several approaches have been proposed to assess the local uncertainty, or confidence, of individual answers by analysis of the semantic answer embeddings. Semantic Density (Qiu & Miikkulainen, 2024) approximates the probability distribution over answers in semantic space and assigns high confidence to answers lying in high probability regions, but uses model likelihoods to do so and as such is not a black-box method. Lin et al. (2024) propose several black-box methods to assess global and local uncertainty, including *Degree* and *Eccentricity*. These methods use graphs computed from pairwise comparisons between answer representations, and as such lose the full semantic context of the response set, which is captured by our method.

**White-box Uncertainty:** Finally, numerous methods have been proposed to estimate uncertainty with white-box model access, using for instance token probabilities, logits or hidden layer activations without requiring additional sampling. These methods include minimum token probability, average token probability, and more advanced techniques that account for the semantic importance of individual tokens (Xia et al., 2025; Zhang et al., 2025; Liu et al., 2024; Malinin & Gales, 2020; Quevedo et al., 2024).

Unlike prior methods that collapse geometry into a single global score, our use of archetypal analysis yields interpretable anchor points that both define batch-level uncertainty and enable principled response-level attribution in a black-box setting. This enables fine-grained hallucination detection and improves interpretability without requiring access to internal model states. We note however that our method uses the semantic dispersion of the sampled responses as a *proxy* for hallucination risk, rather than a guarantee of factual incorrectness.

## 3 Methodology

The proposed geometric framework quantifies LLM uncertainty through a two-tiered analysis of response embeddings. Given a *batch of stochastic samples* generated for a single query, the method first approximates the *geometric support* of the underlying semantic distribution. It then derives two complementary metrics: a *global score* ($H_G$) that quantifies the semantic dispersion of the entire batch to detect hallucinations, and a *local score* ($S$) assigned to *each individual sample* to rank its reliability within the estimated support, enabling correction via preferential answer selection.

### 3.1 Preliminaries and Probabilistic Framework

Let $q$ be an input query to an LLM. We generate a default response $r_{\text{default}}$ by low temperature decoding, and a set of $n$ responses $\{r_1, \ldots, r_n\}$ by sampling with temperature $T > 0$. A sentence embedding model

$\mathcal{E} : \mathcal{S} \to \mathbb{R}^d$ maps each response to an embedding $\mathbf{x}_i = \mathcal{E}(r_i)$. We apply $\ell_2$ normalization and PCA to obtain reduced embeddings $\mathbf{x}_i \in \mathbb{R}^{d'}$, and collect them into $\mathbf{X} \in \mathbb{R}^{n \times d'}$, whose $i$-th row is $\mathbf{x}_i^\top$.

We consider language model outputs to be drawn from a *prompt-conditioned semantic distribution*. For each query $q$ and temperature $T$, the sampling-and-embedding pipeline induces a prompt-conditioned distribution $P_q^{(T)}$ over $\mathbb{R}^{d'}$ such that $\mathbf{x}_i \sim P_q^{(T)}$. We view *semantic uncertainty* as the dispersion of $P_q^{(T)}$, quantified by its (intrinsic) differential entropy. Specifically, if the support of $P_q^{(T)}$ lies in a $d_{\text{int}}$-dimensional affine subspace, we define

$$H(P_q^{(T)}) := - \int p_q^{(T)}(\mathbf{x}) \log p_q^{(T)}(\mathbf{x}) \, d\mathcal{H}^{d_{\text{int}}}(\mathbf{x}), \tag{1}$$

where $\mathcal{H}^{d_{\text{int}}}$ is the $d_{\text{int}}$-dimensional Hausdorff measure restricted to that affine span.

When $P_q^{(T)}$ is supported on a compact set $S_q$ in its $d_{\text{int}}$-dimensional affine span, $H(P_q^{(T)})$ is controlled by the intrinsic $d_{\text{int}}$-volume of $S_q$ ($\mathcal{H}^{d_{\text{int}}}$). Accordingly, our global uncertainty estimator constructs an enclosing *semantic support set* $\widehat{S}_q$ from the finite sample $\mathbf{X}$ and uses $\log \text{Vol}(\widehat{S}_q)$ as a proxy for the semantic entropy $H(P_q^{(T)})$, and hence for global semantic uncertainty.

## 3.2 Geometric Support Estimation via Archetypal Analysis

To estimate the semantic support set $\widehat{S}_q$, we employ Archetypal Analysis (AA) (Cutler & Breiman, 1994). Unlike clustering methods that identify central tendencies (centroids), AA identifies $K$ *archetypes* that lie within the convex hull of the data (typically near its boundary), thereby capturing extremal semantic directions of the model's response set.

Given the response embedding matrix $\mathbf{X} \in \mathbb{R}^{n \times d'}$ (where each row corresponds to a response), AA learns $K$ archetypes that form a dictionary $\mathbf{Z} \in \mathbb{R}^{K \times d'}$. Each response embedding is represented as a convex combination of these archetypes, and each archetype is itself constrained to be a convex combination of the observed embeddings. This yields the bi-convex optimization problem (Abrol & Sharma, 2020):

$$\underset{\substack{\mathbf{B}, \mathbf{A} \\ \mathbf{b}_k \in \Delta_n, \mathbf{a}_i \in \Delta_K}}{\arg\min} \quad \|\mathbf{X} - \mathbf{A}\mathbf{B}\mathbf{X}\|_F^2,$$
$$\Delta_n \triangleq [\mathbf{b}_k \succeq 0, \|\mathbf{b}_k\|_1 = 1], \quad \Delta_K \triangleq [\mathbf{a}_i \succeq 0, \|\mathbf{a}_i\|_1 = 1]. \tag{2}$$

Here, $\mathbf{A} \in \mathbb{R}^{n \times K}$ contains the per-response convex coefficients (row $\mathbf{a}_i^\top$), and $\mathbf{B} \in \mathbb{R}^{K \times n}$ contains the per-archetype convex coefficients (row $\mathbf{b}_k^\top$). The resulting archetypes are given by $\mathbf{Z} = \mathbf{B}\mathbf{X} \in \mathbb{R}^{K \times d'}$.

In practice, Eq. 2 is solved iteratively via block-coordinate descent (Chen et al., 2014), alternating between updates of $\mathbf{A}$ and $\mathbf{B}$ until convergence, yielding archetypes that capture extremal directions of the batch.

ENCLOSURE PROPERTY: The constraints in Eq. 2 ensure that the reconstruction of each response embedding is representable as a convex combination of the learned archetypes. Consequently, the archetypal hull $\widehat{S}_q := \text{conv}(\mathbf{Z})$ encloses the reconstructed batch, and can be interpreted as an estimated semantic support set for the sampled responses up to the AA reconstruction residual (which is minimized by the objective). We formalize this property in Lemma 1.

**Lemma 1** (AA yields an enclosing convex set). *Let $\mathbf{Z} = \{\mathbf{z}_1, \ldots, \mathbf{z}_K\}$ be the archetypes learned by AA, and let $\boldsymbol{\alpha}_i \in \Delta_K$ denote the coefficients reconstructing $\hat{\mathbf{x}}_i$. Then every reconstructed response satisfies $\hat{\mathbf{x}}_i \in \text{conv}(\mathbf{Z})$.*

*Proof.* By construction, $\hat{\mathbf{x}}_i = \sum_{k=1}^K \alpha_{ik} \mathbf{z}_k$ with $\alpha_{ik} \geq 0$ and $\sum_{k=1}^K \alpha_{ik} = 1$. Hence $\hat{\mathbf{x}}_i$ is a convex combination of archetypes and lies in $\text{conv}(\mathbf{Z})$. $\square$

## 3.3 Global Uncertainty via Geometric Volume

We define global uncertainty based on the size of the estimated semantic support set $\widehat{S}_q = \text{conv}(\mathbf{Z})$. A high degree of uncertainty manifests as a diverse set of responses, which geometrically corresponds to archetypes

that are far apart in the embedding space. We capture this dispersion by computing the intrinsic volume of the archetypal hull. Our global metric, *Geometric Volume*, is the logarithm of this volume:

$$H_G(\mathbf{X}) = \frac{1}{d_{\text{int}}} \log \left( \text{Vol}(\widehat{S}_q) + \epsilon \right), \tag{3}$$

where $\epsilon$ is a small constant (e.g., $10^{-12}$) for numerical stability. To ensure robustness across queries, we normalize by the intrinsic dimension $d_{\text{int}}$ of the hull.

If the number of archetypes $K$ is small ($K - 1 < d'$), the hull is degenerate in $\mathbb{R}^{d'}$ (zero volume); in such cases, we compute the volume within the $(K-1)$-dimensional affine span of the archetypes.

### 3.3.1 Theoretical Justification

The rationale to employ *Geometric Volume* as a measure of uncertainty is grounded in information theory. As discussed earlier, the spatial size or volume of a support set fundamentally limits the entropy of any distribution defined over it. Intuitively, a larger hull offers more 'room' for the semantic distribution to spread out. Conversely, a small volume forces the distribution to be concentrated, associated with low uncertainty. Therefore, by measuring volume, we are measuring the *capacity* for uncertainty.

We formalise this relationship in Theorem 1, which establishes that the log-volume is a strict upper bound on differential entropy. To address the fact that the semantic distribution $P_q$ is supported on the archetypal hull - a lower-dimensional simplex embedded in a high-dimensional space - we formulate the bound using the Hausdorff measure on the affine span.

**Theorem 1.** *Let $\mathcal{Z} = \{\mathbf{z}_1, \ldots, \mathbf{z}_K\} \subset \mathbb{R}^{d'}$ be a set of $K$ affinely independent archetypes. Let $\Delta = \text{conv}(\mathcal{Z})$ denote the $(K-1)$-dimensional simplex they span, with intrinsic volume $V > 0$ measured using the $(K-1)$-dimensional Hausdorff measure $\mathcal{H}^{K-1}$ on the affine span of $\Delta$. Let $\mathbf{x}$ be a continuous random vector supported on $\Delta$, with density $p(\mathbf{x})$ that is absolutely continuous with respect to $\mathcal{H}^{K-1}$. Then the differential entropy of $\mathbf{x}$ satisfies:*

$$H(\mathbf{x}) = -\int_\Delta p(\mathbf{x}) \log p(\mathbf{x}) \, d\mathcal{H}^{K-1}(\mathbf{x}) \leq \log V, \tag{4}$$

*and the upper bound is achieved if and only if $\mathbf{x}$ is uniformly distributed over $\Delta$.*

*Proof.* See Appendix B.1. □

Theorem 1 upper-bounds the entropy of any distribution supported on a set by the log-volume of that set. Since the true semantic support $S_q$ of $P_q$ is unknown, we estimate it from the finite sample $\mathbf{X}$ via the archetypal hull $\widehat{S}_q = \text{conv}(\mathbf{Z})$. The next proposition applies Theorem 1 to this estimated support, justifying $H_G$ as a proxy for semantic uncertainty.

**Proposition 1** (Volume as an entropy upper bound). *Assume the semantic distribution $P_q$ is (approximately) supported on the archetypal hull $\widehat{S}_q$ (i.e., $S_q$ is well-approximated by $\widehat{S}_q$). Then the semantic entropy is bounded by the geometric volume:*

$$H(P_q) \lesssim \log \text{Vol}(\widehat{S}_q). \tag{5}$$

*Consequently, $H_G(\mathbf{X})$ in equation 3 serves as a theoretically grounded proxy for the worst-case semantic uncertainty of the model.*

*Proof.* See Appendix B.2. □

By framing Geometric Volume as an entropy upper bound, we elevate it from a heuristic notion of geometric dispersion to a principled proxy for prompt-conditioned semantic uncertainty. In practice, a large $H_G(\mathbf{X})$ indicates that the sampled responses occupy a high-capacity semantic region, consistent with a broad (high-entropy) semantic distribution, and thus flags queries for which the model expresses substantial uncertainty. We stress however that Proposition 1 is a statement about the entropy of the embedded response distribution: the log-volume upper-bounds the semantic entropy only to the extent that the archetypal hull $\widehat{S}_q$ faithfully approximates the true support $S_q$.

### 3.3.2 Support Geometry and Tightness of AA Volume

To ensure the geometric volume metric is both computable and precise, we now formalize the geometry of the archetypal support. We first derive a closed-form expression for the simplex volume, and then demonstrate that relaxing simplex constraints to box constraints - a characteristic of comparative methods - substantially inflates the measured volume, yielding a provably looser uncertainty proxy.

In the affinely independent case, the archetypal hull $\widehat{S}_q = \mathrm{conv}(\mathbf{Z})$ is a simplex of intrinsic dimension $d_{\mathrm{int}} = K - 1$. The following lemma provides an explicit Gram-determinant formula for its intrinsic volume.

**Lemma 2** (Simplex volume as a Gram determinant). *Let $\mathcal{Z} = \{\mathbf{z}_1, \ldots, \mathbf{z}_K\} \subset \mathbb{R}^{d'}$ be affinely independent and $m := K - 1$. Define the edge matrix $E = [\, \mathbf{z}_2 - \mathbf{z}_1, \ldots, \mathbf{z}_K - \mathbf{z}_1 \,] \in \mathbb{R}^{d' \times m}$. Then the intrinsic $m$-volume of $\Delta = \mathrm{conv}(\mathcal{Z})$ satisfies*

$$\mathrm{Vol}_m(\Delta) = \frac{1}{m!}\sqrt{\det(E^\top E)}. \tag{6}$$

*Proof.* See Appendix B.3 $\qquad\qquad\square$

Lemma 2 shows that the AA support volume admits a translation-invariant Gram form on *edge vectors*. More generally, dispersion-based 'volume' estimators can be interpreted as measuring the size of a feasible region under different coefficient constraints. To illustrate the tightness of our convex hull estimator compared to the estimator used by Li et al. (2025) for Semantic Volume, we consider the volume of the enclosing parallelepiped. Replacing the simplex constraints $\{\mathbf{u} \succeq 0, \ \mathbf{1}^\top \mathbf{u} \leq 1\}$ with the looser box constraint $\mathbf{u} \in [0,1]^m$ yields a parallelepiped outer approximation.

**Proposition 2** (Parallelepiped–simplex factorial gap). *Let $\mathcal{Z} = \{\mathbf{z}_1, \ldots, \mathbf{z}_K\} \subset \mathbb{R}^{d'}$ be affinely independent and set $m := K - 1$. Fix $\mathbf{z}_1$ as a reference vertex and define the edge matrix*

$$E := [\, \mathbf{z}_2 - \mathbf{z}_1, \ \mathbf{z}_3 - \mathbf{z}_1, \ \ldots, \ \mathbf{z}_K - \mathbf{z}_1 \,] \in \mathbb{R}^{d' \times m}. \tag{7}$$

*Define the simplex support in edge coordinates,*

$$\mathcal{S} := \{E\mathbf{u} : \ \mathbf{u} \succeq 0, \ \mathbf{1}^\top \mathbf{u} \leq 1\}, \tag{8}$$

*and its parallelepiped (box-constrained) outer approximation,*

$$\mathcal{P} := \{E\mathbf{u} : \ \mathbf{u} \in [0,1]^m\}. \tag{9}$$

*Then $\mathcal{S} \subseteq \mathcal{P}$ and their intrinsic $m$-volumes satisfy*

$$\mathrm{Vol}_m(\mathcal{P}) = m! \, \mathrm{Vol}_m(\mathcal{S}). \tag{10}$$

*Proof.* See Appendix B.3. $\qquad\qquad\square$

The factor $m!$ exposes a sharp geometric gap between the simplex support and its box-constrained outer approximation. As the intrinsic dimension $m$ grows, this ratio increases factorially, so the enclosing parallelepiped allocates most of its volume to regions that are infeasible under the simplex coefficient constraints. By computing the intrinsic volume of the simplex exactly, $H_G$ avoids counting this extraneous volume, preventing the systematic inflation of uncertainty that arises from bounding-box style outer approximations.

### 3.4 Local Uncertainty via Geometric Suspicion

Given a query $q$, let $\{r_i\}_{i=1}^n$ denote $n$ stochastic responses sampled for $q$, and let $\{\mathbf{x}_i\}_{i=1}^n \subset \mathbb{R}^{d'}$ be their embeddings (after the same preprocessing described in Section 3.1). We assign each response a *local suspicion* score $S(r_i)$ that ranks samples within $\{r_i\}_{i=1}^n$ by geometric atypicality. The guiding principles are that hallucinated samples are more likely to be *(i) locally isolated, (ii) far from the dominant semantic consensus,* or *(iii) explained by rare/extremal archetypal directions.* We instantiate these criteria via three complementary metrics and use a combined score to rank samples, as described in Figure 2.

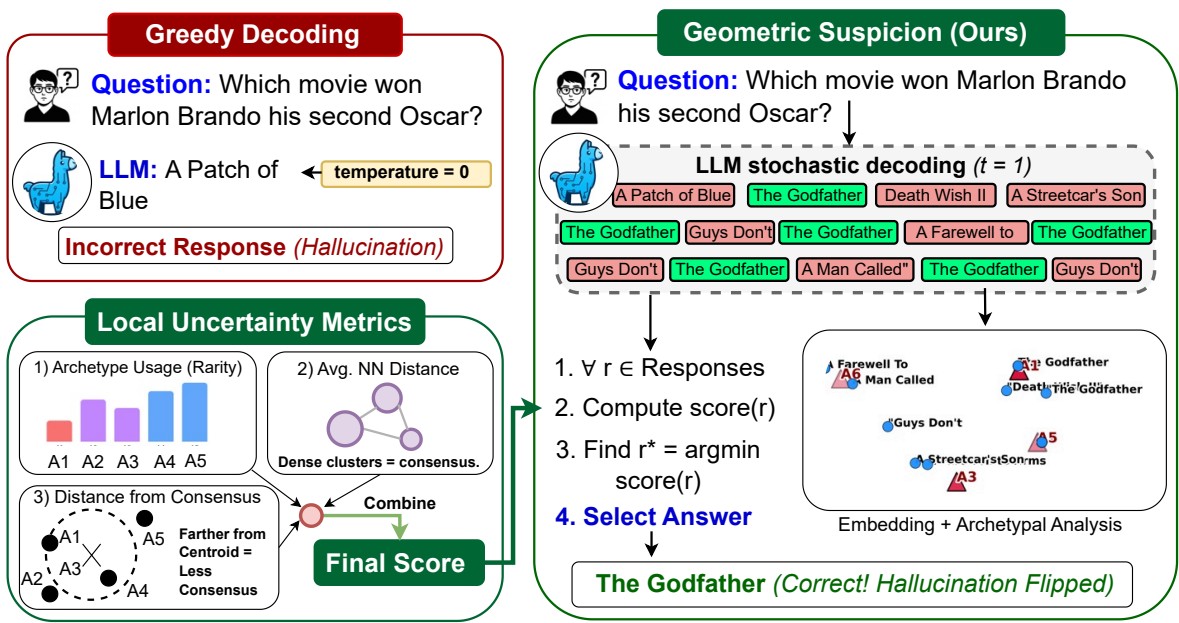

Figure 2: We employ our composite metric Geometric Suspicion to correct hallucinations in a preferential selection framework.

We introduce the metrics and a principled method for their combination below. The first two metrics align closely with established local uncertainty baselines, namely *Degree* and *Eccentricity* (Lin et al., 2024). These baselines rely on an answer-answer affinity graph to determine the relative confidence of answers within a batch. *Degree* is a monotone transform of local similarity mass, whilst *Eccentricity* penalizes globally off-centre responses; the proven effectiveness of both approaches supports our first two guiding principles. The final metric is uniquely grounded in our archetypal framework, and empirical analysis of our generated datasets shows its utility for hallucination mitigation. Our approach builds on existing work by treating these confidence measures as complementary rather than competing signals.

(1) LOCAL SPARSITY: Let $N_k(\mathbf{x}_i)$ be the set of $k$ nearest neighbours of $\mathbf{x}_i$ among $\{\mathbf{x}_j\}_{j \neq i}$ (under the same distance used throughout). We define

$$L(r_i) \ := \ \frac{1}{k} \sum_{\mathbf{x}_j \in N_k(\mathbf{x}_i)} \|\mathbf{x}_i - \mathbf{x}_j\|_2^2. \tag{11}$$

Large $L(r_i)$ indicates that $r_i$ lies in a *locally sparse* region of the empirical support induced by the sampled responses. This metric is analogous to *Degree* in previous graph-based approaches.

(2) DISTANCE FROM CONSENSUS: To place each response in a global geometric context, we measure the distance to the batch consensus

$$D(r_i) = \left\|\mathbf{x}_i - \mathbf{x}_c\right\|_2, \qquad \mathbf{x}_c = \frac{1}{n} \sum_{j=1}^{n} \mathbf{x}_j. \tag{12}$$

Large values indicate semantic deviation from the central tendency of the answer set. This metric is analogous to *Eccentricity* in prior graph methods.

(3) ARCHETYPAL USAGE RARITY: The $i$-th row of $\mathbf{A}$, denoted $\boldsymbol{\alpha}_i = [A_{i1}, \ldots, A_{iK}]$ contains coefficients describing how the response embedding $\mathbf{x}_i$ is reconstructed from the set of learned archetypes $\mathbf{Z}$. The score for a response $r_i$ is high if its reconstruction coefficients $\mathbf{A}_{ik}$ depend heavily on archetypes that are, on

average, rarely used to explain the rest of the data:

$$U(r_i) = \sum_{k=1}^{K} A_{ik}\big(1 - \bar{\alpha}_k\big), \quad \text{where} \quad \bar{\alpha}_k = \frac{1}{n}\sum_{j=1}^{n} A_{jk}. \tag{13}$$

Usage rarity connects our local analysis directly to the global *Geometric Volume* measure. Since the archetypes define the vertices of the enclosing semantic convex hull (Lemma 1), a response with high usage rarity is one that relies heavily on a vertex that is otherwise utilized negligibly by the batch. Geometrically, this implies the response occupies a sparse region near the boundary of the simplex, far from the semantic consensus. Such responses effectively expand the geometric support, thereby inflating the volume and the corresponding upper bound on the semantic distribution's entropy.

**Nonparametric extremeness and aggregation.** To place the three scores on a common scale, we convert each into a within-set right-tail extremeness score using an empirical p-value:

$$p_T(r_i) := \frac{1 + \sum_{j=1}^{n} \mathbb{I}[T(r_j) \ge T(r_i)]}{n + 1}, \qquad T \in \{L, D, U\}. \tag{14}$$

Finally, we aggregate extremeness across criteria via Fisher's method:

$$S(r_i) := -2 \sum_{T \in \{L,D,U\}} \log p_T(r_i). \tag{15}$$

We use $S(r_i)$ as a *ranking* score; we do not interpret it as a calibrated p-value, since the component scores can be dependent.

### 3.4.1 Theoretical Justification: Distribution-Free Extremeness and Dependence-Robust Aggregation

We now provide a minimal mathematical grounding for the score $S(r_i)$ in equation 15. The key design choice is to (i) convert the heterogeneous geometric statistics $T \in \{L, D, U\}$ into *rank-based* right-tail extremeness scores $p_T(r_i)$ via equation 14, and then (ii) aggregate them using Fisher's transform. This yields two desirable properties: the normalization is *distribution-free* under a weak symmetry assumption (exchangeability), and the aggregation admits an explicit *dependence-robust* extremeness guarantee. Consequently, for any fixed $i$, large values of $S(r_i)$ remain provably rare under a null in which no sample is atypical.

EXCHANGEABLE NULL: As a modelling lens, we consider a 'no-anomaly' null under which the sampled responses $\{r_i\}_{i=1}^{n}$ (equivalently, the induced scores) for query $q$ are *exchangeable*. Informally, the index $i$ carries no information, so any sample is equally likely to be the $i$-th draw. This is the natural null for within-set ranking: if no response is atypical, its rank among $\{r_j\}$ should behave like a random rank.

Under this 'no-anomaly' null, the exchangeability of the sampled responses $\{r_i\}_{i=1}^{n}$ ensures that the resulting score vectors are also exchangeable. Specifically, because the scoring functions $L$, $D$, and $U$ are *permutation-equivariant*—treating the input set as an unordered collection—the induced scores $(T(r_1), \dots, T(r_n))$ preserve the exchangeable property for any $T \in \{L, D, U\}$. Building on this symmetry, the following lemma demonstrates that the empirical construction in equation 14 yields valid, conservative extremeness scores across all three criteria.

**Lemma 3** (Validity of rank-based right-tail extremeness). *Let $T \in \{L, D, U\}$ and define $p_T(r_i)$ as in equation 14. If $(T(r_1), \dots, T(r_n))$ is exchangeable (ties allowed), then for all $t \in [0, 1]$,*

$$\mathbb{P}(p_T(r_i) \le t) \ \le \ t. \tag{16}$$

*Proof.* See Appendix B.4. □

Lemma 3 justifies equation 14 as a principled, non-parametric normalisation. Under exchangeability, $p_T(r_i)$ is super-uniform (and thus a valid, typically conservative, right-tail p-value for extremeness), so thresholding

at level $t$ flags a non-extreme sample with probability at most $t$. It converts each geometric statistic into a common extremeness scale without assuming any parametric form for embedding distributions or any calibration of the raw scores.

Having placed the component criteria on a common extremeness scale, we can justify the aggregation step. The following proposition shows that Fisher's transform yields an omnibus score with an explicit tail bound even when the component p-values are arbitrarily dependent.

**Proposition 3** (Dependence-robust extremeness guarantee for Fisher aggregation)**.** *Define $S(r_i)$ by equation 15. Suppose each component p-value $\{p_T(r_i)\}_{T \in \{L,D,U\}}$ is super-uniform (e.g., by Lemma 3 under exchangeability), while allowing arbitrary dependence between them. Then for all $s \geq 0$,*

$$\mathbb{P}(S(r_i) \geq s) \;\leq\; 3 \exp\!\left(-\frac{s}{6}\right). \tag{17}$$

*Equivalently, the mapping*

$$p_{\mathrm{comb}}(r_i) := \min\!\left\{1,\; 3\exp\!\big(-S(r_i)/6\big)\right\} \tag{18}$$

*defines a valid (conservative) combined p-value, i.e. for all $t \in [0,1]$, $\mathbb{P}(p_{\mathrm{comb}}(r_i) \leq t) \leq t$, under the exchangeable null.*

*Proof.* See Appendix B.4. $\square$

Proposition 3 formalizes the role of equation 15 as an omnibus *extremeness aggregator*. Because $L$, $D$, and $U$ are computed from the same sampled set, the component p-values can be strongly dependent, so we do not treat $S(r_i)$ as a classically calibrated Fisher test statistic. Nevertheless, equation 17 guarantees that large values of $S(r_i)$ are still provably rare under the exchangeable null. The mapping in equation 18 converts this tail bound into a conservative combined p-value; in practice, we use $S(r_i)$ primarily as a ranking statistic.

To further support our specific choice of terms, we conduct an empirical analysis of the datasets generated for our global uncertainty experiments, which can be found in Appendix Section E. In Figure 4, we show examples where component metrics used in isolation would not select an optimal response from a batch, whereas their combination overcomes individual limitations. Density-based measures, for instance, struggle in situations where there exist multiple dense clusters in the semantic response set.

## 4 Experiments

We conduct a series of experiments to evaluate the effectiveness of our geometric framework for hallucination detection and correction. We assess its performance under external uncertainty (prompt ambiguity) and internal uncertainty (fact-checking). We also evaluate uncertainty methods in more realistic, long-form medical question-answering scenarios that better reflect real-world conditions.

### 4.1 Benchmarks and Baselines

We evaluate on five benchmarks covering both external and internal uncertainty: CLAMBER (ambiguous prompts), TriviaQA (short-form QA), ScienceQA (scientific reasoning), MedicalQA (high-stakes medical QA), and K-QA (real-world medical scenarios). These benchmarks span ambiguous queries, factual QA, and long-form medical domains. Dataset construction details are provided in Section C of the Appendix.

For global uncertainty, we compare our method with Semantic Volume (Li et al., 2025), Semantic Entropy (Farquhar et al., 2024), and P(true), which estimates the probability that a model's generated response is correct by directly asking the model itself to self-assess its confidence (Kadavath et al., 2022; Lin et al., 2024). For local uncertainty, we assess the *Geometric Suspicion* score in reducing hallucinations via a Best-of-N selection strategy. We compare our method with Degree and Eccentricity (Lin et al., 2024), which are based on graphs derived from pairwise response comparisons. Each experiment is repeated three times to mitigate sampling stochasticity, and we report the mean and standard deviation of all metrics.

We selected baselines that operate in the same setting as our method (black-box access only, with no model weights or gradients) and that together span differing families of sampling-based uncertainty estimation: semantic clustering (Semantic Entropy), geometric dispersion-based (Semantic Volume), self-evaluation (P(true)), and graph-based local measures (Degree and Eccentricity).

## 4.2 Hallucination Detection via Global Uncertainty

Our Geometric Volume metric can be used to detect hallucinations at the prompt level. Given $r_{\text{default}}$ and a corresponding set of responses with embedding matrix $\mathbf{X}$, we classify $r_{\text{default}}$ as hallucinated if the global uncertainty exceeds a threshold $\tau$, i.e., if $H_G(\mathbf{X}) > \tau$. The threshold $\tau$ is selected on a small labelled validation split (a 10% subset) by maximising F1. We emphasize that while the Geometric Volume score requires no supervision, this thresholding step does, hence the detector is not calibration-free in deployment.

Geometric Volume demonstrates strong performance across all benchmarks and models investigated, with results shown in Table 1. On K-QA, it significantly outperforms baselines, achieving the best F1 Score and AUROC with GPT-3.5-Turbo (75.4 and 67.8) and the strongest AUROC with GPT-4o Mini (67.7) and Llama3.1-8b (72.8). On MedicalQA, it remains highly competitive, delivering top F1 Scores with GPT-3.5-Turbo (73.9) and Qwen3-8b (75.2), as well as the best AUROC with GPT-4o Mini (60.9).

On CLAMBER, which tests detection of ambiguous prompts, Geometric Volume achieves the highest AUROC for GPT-4o Mini (61.7) and Qwen3-8b (59.1), and the highest F1 scores for GPT-3.5-Turbo and Llama3.1-8b. In TriviaQA, it results in the highest F1 and AUROC with GPT-4o Mini (71.1, 76.7) and GPT-3.5-Turbo (71.8 F1, 76.7 AUROC). On ScienceQA, it shows particular strength with Llama3.1-8b, delivering 78.6 F1 and 81.6 AUROC. These results highlight its ability to model semantic dispersion effectively.

## 4.3 Hallucination Correction via Local Uncertainty

Beyond merely identifying uncertain responses, the primary application of our suspicion score $S(r)$ is to actively improve model reliability. We employ it in a Best-of-N (BoN) sampling framework, where the goal is to select the most plausible response from a set of $n$ candidates generated for a given question. We replace the model's default (and potentially hallucinated) answer with the batch response $r^*$ that exhibits the minimum suspicion score.

To quantify the efficacy of this selection strategy, we measure the net reduction in the hallucination rate. First, we define the baseline hallucination rate, $H_{\text{baseline}}$, as the proportion of questions in a test set for which the model's default answer $r_{\text{default}}$ was a hallucination. Next, we define the post-selection hallucination rate, $H_{\text{BoN-U}}$, as the proportion of questions where the answer $r^*$ selected by our uncertainty-guided method is a hallucination. The improvement is captured by the rate change $\Delta H$:

$$\Delta H = H_{\text{baseline}} - H_{\text{BoN-U}}. \tag{19}$$

We evaluate our local UQ methods on the same datasets curated for the global internal uncertainty experiments. We first filter each dataset to only contain questions where the set of sampled responses contains examples of both hallucinations and non-hallucinations. For each sample in the perturbed responses, we then compute the local uncertainty score using the archetypal reconstruction matrix $\mathbf{A}$ and embeddings $\mathbf{X}$. We report $\Delta H$ and AUARC (Area Under the Accuracy–Rejection Curve). The latter is computed per question by ranking responses by uncertainty, progressively rejecting the top-$k$, measuring accuracy on the retained set, and averaging over all rejection rates ($k = 0, \ldots, n - 1$). A random predictor yields AUARC equal to the base accuracy.

We show in Table 2 that using our local uncertainty metric, we are able to reduce hallucination rates across nearly all models and datasets investigated. In general it surpasses baselines, with particularly dominant performance in the MedicalQA task. The sole exception where our metric is unable to reduce the hallucination rate is the Qwen3-8B model on the TriviaQA dataset, where all baselines fail similarly. We find that compared to other model and dataset combinations, in this case the model is more often *confidently wrong* when it hallucinates: that is, very few of the sampled answers are non-hallucinations, so choosing an

Table 1: Results across five benchmarks and four LLMs comparing global uncertainty estimation methods. Reported values are mean$_{\text{std}}$. Blue cells indicate best results from **Ours**, red cells indicate best results from baselines. AUC = AUROC.

| Method | CLAMBER | | TriviaQA | | ScienceQA | | MedicalQA | | K-QA | |
|---|---|---|---|---|---|---|---|---|---|---|
| | F1 | AUC | F1 | AUC | F1 | AUC | F1 | AUC | F1 | AUC |
| **GPT-4o Mini** | | | | | | | | | | |
| $p(\text{true})$ | $47.0_{1.6}$ | $58.5_{0.4}$ | $59.2_{3.6}$ | $65.8_{4.0}$ | $53.0_{4.0}$ | $77.6_{3.4}$ | $73.6_{1.2}$ | $59.6_{1.0}$ | $61.5_{4.7}$ | $63.4_{1.5}$ |
| Sem. Entropy | $63.9_{0.3}$ | $48.1_{0.8}$ | $67.7_{3.4}$ | $75.3_{0.3}$ | $48.6_{2.7}$ | $65.3_{1.6}$ | $70.9_{1.0}$ | $59.3_{1.0}$ | $60.4_{4.8}$ | $50.5_{5.7}$ |
| Sem. Volume | $68.5_{0.4}$ | $55.8_{0.9}$ | $69.9_{1.0}$ | $74.4_{0.3}$ | $54.0_{4.4}$ | $57.9_{1.5}$ | $73.0_{1.2}$ | $59.4_{1.0}$ | $68.2_{1.6}$ | $65.2_{1.1}$ |
| **Ours** | $66.1_{0.3}$ | $61.7_{0.7}$ | $71.1_{1.4}$ | $76.7_{0.3}$ | $58.7_{3.0}$ | $68.8_{2.2}$ | $73.6_{1.2}$ | $60.9_{1.4}$ | $68.6_{0.8}$ | $67.7_{1.3}$ |
| **GPT-3.5-Turbo** | | | | | | | | | | |
| $p(\text{true})$ | $53.0_{0.5}$ | $60.1_{0.6}$ | $49.6_{1.0}$ | $68.2_{2.7}$ | $57.0_{5.6}$ | $76.1_{1.5}$ | $60.0_{11.1}$ | $71.9_{1.9}$ | $71.8_{1.4}$ | $60.5_{1.2}$ |
| Sem. Entropy | $66.6_{0.2}$ | $53.8_{0.2}$ | $69.6_{1.2}$ | $75.6_{0.4}$ | $62.3_{3.3}$ | $69.2_{1.5}$ | $71.3_{1.5}$ | $68.7_{0.3}$ | $40.3_{6.9}$ | $66.4_{1.2}$ |
| Sem. Volume | $68.5_{0.1}$ | $54.4_{0.6}$ | $71.7_{0.6}$ | $74.9_{0.8}$ | $64.3_{3.2}$ | $62.7_{4.5}$ | $73.1_{0.3}$ | $61.7_{0.6}$ | $74.8_{1.0}$ | $65.3_{0.7}$ |
| **Ours** | $82.5_{2.6}$ | $61.2_{1.3}$ | $71.8_{1.5}$ | $76.7_{0.4}$ | $70.1_{3.7}$ | $73.4_{3.6}$ | $73.9_{1.1}$ | $64.9_{0.4}$ | $75.4_{0.4}$ | $67.8_{3.2}$ |
| **Qwen3-8b** | | | | | | | | | | |
| $p(\text{true})$ | $55.0_{0.8}$ | $57.1_{0.3}$ | $64.1_{3.6}$ | $86.0_{1.1}$ | $41.3_{4.3}$ | $66.6_{1.2}$ | $74.4_{1.3}$ | $66.9_{1.7}$ | $72.8_{0.8}$ | $66.3_{1.4}$ |
| Sem. Entropy | $63.8_{0.2}$ | $49.3_{0.5}$ | $78.1_{0.6}$ | $84.2_{0.3}$ | $22.2_{19.3}$ | $58.9_{0.8}$ | $73.2_{0.5}$ | $71.1_{0.6}$ | $73.4_{1.1}$ | $72.5_{0.8}$ |
| Sem. Volume | $67.1_{0.2}$ | $55.3_{0.2}$ | $74.4_{1.4}$ | $80.8_{0.7}$ | $66.6_{0.1}$ | $48.1_{0.6}$ | $74.4_{0.8}$ | $69.1_{0.3}$ | $73.3_{1.1}$ | $69.8_{0.2}$ |
| **Ours** | $65.9_{0.1}$ | $59.1_{0.4}$ | $76.9_{0.3}$ | $82.1_{0.4}$ | $66.7_{0.2}$ | $67.2_{0.4}$ | $75.2_{1.1}$ | $69.5_{0.6}$ | $73.6_{1.2}$ | $69.8_{0.2}$ |
| **Llama3.1-8b** | | | | | | | | | | |
| $p(\text{true})$ | $62.8_{2.0}$ | $62.0_{0.5}$ | $60.8_{2.7}$ | $58.2_{12.2}$ | $45.3_{2.9}$ | $69.1_{2.7}$ | $74.4_{2.2}$ | $76.4_{2.1}$ | $55.8_{3.8}$ | $67.9_{4.0}$ |
| Sem. Entropy | $67.1_{0.4}$ | $58.0_{0.7}$ | $71.0_{1.0}$ | $76.9_{0.6}$ | $73.0_{0.8}$ | $77.0_{0.8}$ | $76.6_{0.7}$ | $72.0_{1.2}$ | $82.8_{1.9}$ | $69.9_{2.5}$ |
| Sem. Volume | $66.8_{0.1}$ | $51.0_{0.4}$ | $70.6_{0.0}$ | $72.7_{0.7}$ | $66.4_{0.2}$ | $75.4_{0.7}$ | $75.4_{0.2}$ | $56.2_{0.7}$ | $81.7_{0.4}$ | $65.0_{2.2}$ |
| **Ours** | $76.2_{2.1}$ | $54.9_{1.2}$ | $71.3_{0.5}$ | $75.4_{0.5}$ | $78.6_{0.7}$ | $81.6_{0.4}$ | $75.3_{0.2}$ | $67.2_{0.9}$ | $82.7_{0.1}$ | $72.8_{1.1}$ |

optimal answer becomes a difficult task. We comment on this method limitation in Section 5 and provide the evidence for this conclusion in the following section.

### 4.3.1 Local Uncertainty Efficacy

We next quantify our claim that the efficacy of local uncertainty measures is inherently bounded by the quality of the sampled response set. We analyzed the relationship between the reduction in hallucination rate ($\Delta H$) and the proportion of correct answers available in the sampled batch for questions where the default response was a hallucination.

Figure 3 illustrates this relationship across all model and dataset combinations. The x-axis represents the *potential to correct*, defined as the median proportion of non-hallucinated answers found in the sampled batch given that the default answer was incorrect. The y-axis shows the actual performance gain ($\Delta H$). We observe a strong positive correlation. Notably, the case of Qwen3-8B on TriviaQA (the lowest performance in $\Delta H$ terms) demonstrates the 'confidently wrong' failure mode: while the model's default answer is incorrect, the sampled batch also contains very few correct alternatives. In such regimes, the semantic space is collapsed around the hallucination, and no re-ranking method can be effective.

### 4.3.2 Qualitative Analysis

In Figure 4 we present examples demonstrating the utility of our composite local score. Figure 4(a) shows a straightforward example where all three of the component terms have high p-values. The correct answer is in a dense local cluster, near a global consensus point, and can be reconstructed using commonly used

Table 2: Comparison of local uncertainty methods on ΔH (left) and AUARC (right). Values are percentages; subscripts denote standard deviation across runs. Blue cells indicate the best value among **Ours**; red cells indicate the best baseline.

| | TriviaQA | | ScienceQA | | MedicalQA | | K-QA | |
|---|---|---|---|---|---|---|---|---|
| **Method** | ΔH | AUARC | ΔH | AUARC | ΔH | AUARC | ΔH | AUARC |
| **GPT-4o Mini** | | | | | | | | |
| Eccentricity | $-1.0_{1.1}$ | $48.3_{1.0}$ | $4.9_{0.9}$ | $16.4_{4.9}$ | $2.7_{1.0}$ | $55.7_{2.1}$ | $4.9_{4.0}$ | $58.5_{0.2}$ |
| Degree | $7.7_{2.5}$ | $51.9_{0.3}$ | $5.7_{0.6}$ | $16.8_{4.6}$ | $7.0_{3.7}$ | $57.8_{2.4}$ | $15.1_{1.9}$ | $63.7_{0.3}$ |
| **Ours** | $6.9_{1.1}$ | $51.9_{0.2}$ | $6.2_{1.0}$ | $17.3_{4.8}$ | $8.9_{1.6}$ | $58.6_{1.8}$ | $15.3_{2.9}$ | $64.1_{0.2}$ |
| **GPT-3.5-Turbo** | | | | | | | | |
| Eccentricity | $1.0_{2.0}$ | $55.8_{0.2}$ | $5.5_{1.4}$ | $25.8_{5.7}$ | $1.7_{2.0}$ | $51.5_{1.2}$ | $20.8_{4.5}$ | $64.3_{0.2}$ |
| Degree | $5.7_{0.8}$ | $58.1_{1.0}$ | $4.5_{2.9}$ | $25.1_{6.6}$ | $8.7_{1.4}$ | $54.4_{0.6}$ | $32.2_{3.2}$ | $68.8_{1.6}$ |
| **Ours** | $6.0_{0.6}$ | $58.1_{0.7}$ | $2.8_{2.5}$ | $24.5_{6.8}$ | $8.8_{0.6}$ | $55.0_{0.5}$ | $31.3_{3.0}$ | $68.3_{1.7}$ |
| **Qwen3-8b** | | | | | | | | |
| Eccentricity | $-9.2_{1.8}$ | $47.3_{0.9}$ | $2.2_{2.3}$ | $13.2_{0.9}$ | $-1.0_{1.6}$ | $49.7_{2.9}$ | $9.8_{4.8}$ | $55.2_{2.8}$ |
| Degree | $-0.7_{1.6}$ | $51.6_{0.8}$ | $3.7_{3.3}$ | $13.7_{0.1}$ | $7.5_{2.2}$ | $53.4_{2.3}$ | $15.3_{8.6}$ | $58.5_{3.6}$ |
| **Ours** | $-0.5_{0.7}$ | $51.6_{0.8}$ | $3.7_{3.3}$ | $14.8_{0.5}$ | $8.0_{1.1}$ | $54.1_{2.4}$ | $19.0_{7.2}$ | $59.9_{3.2}$ |
| **Llama3.1-8b** | | | | | | | | |
| Eccentricity | $-2.6_{0.3}$ | $42.2_{0.2}$ | $8.2_{1.8}$ | $25.0_{1.6}$ | $-7.5_{2.9}$ | $41.8_{0.8}$ | $9.2_{3.9}$ | $44.8_{0.6}$ |
| Degree | $4.5_{2.1}$ | $45.5_{0.8}$ | $7.0_{3.0}$ | $24.5_{2.1}$ | $4.0_{3.0}$ | $47.2_{0.8}$ | $12.3_{3.3}$ | $46.4_{1.6}$ |
| **Ours** | $2.1_{1.7}$ | $44.7_{0.6}$ | $8.0_{2.9}$ | $25.5_{2.0}$ | $5.2_{1.2}$ | $48.6_{1.3}$ | $15.3_{3.6}$ | $46.9_{1.3}$ |

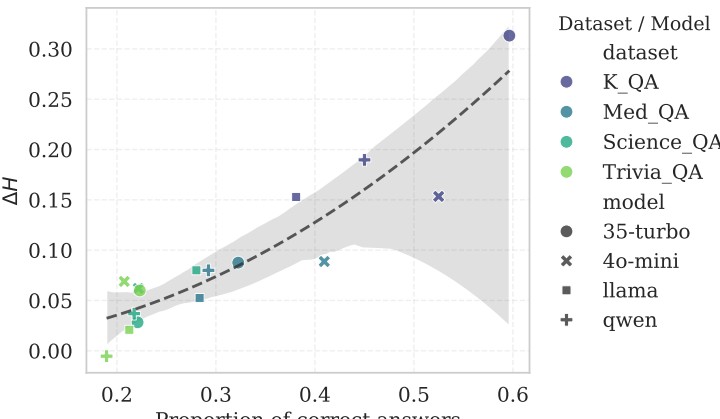

Figure 3: **Geometric suspicion utility vs. corrective potential.** The reduction in hallucination rate $(\Delta H)$ is plotted against the median proportion of correct answers found in the sampled batch for instances where the default answer was incorrect. The dashed curve represents a second-order polynomial regression fit highlighting the general scaling trend, while the shaded region indicates the 95% confidence interval. Performance naturally drops in 'confidently wrong' regimes where the sampled batch lacks correct alternatives, but improves as the corrective potential increases.

archetypes. Figure 4(b) shows a slightly harder example, where there are multiple clusters of semantically similar answers. Here the desired answer is not within the densest local cluster or the closest to a global consensus, but has lower suspicion due to its non-reliance on rarely used archetypes. The example in Figure 4(c) can be used to understand the necessity of both local density and distance from consensus metrics. Here there exist multiple dense clusters, resulting in high local density for the majority of the batch. The global semantic consensus point however is closer to the 'Indonesia' cluster than it is the 'not specified' or 'unknown entity' clusters; the hallucinated answers of 'Brazil' or 'Pakistan' are countries and semantically

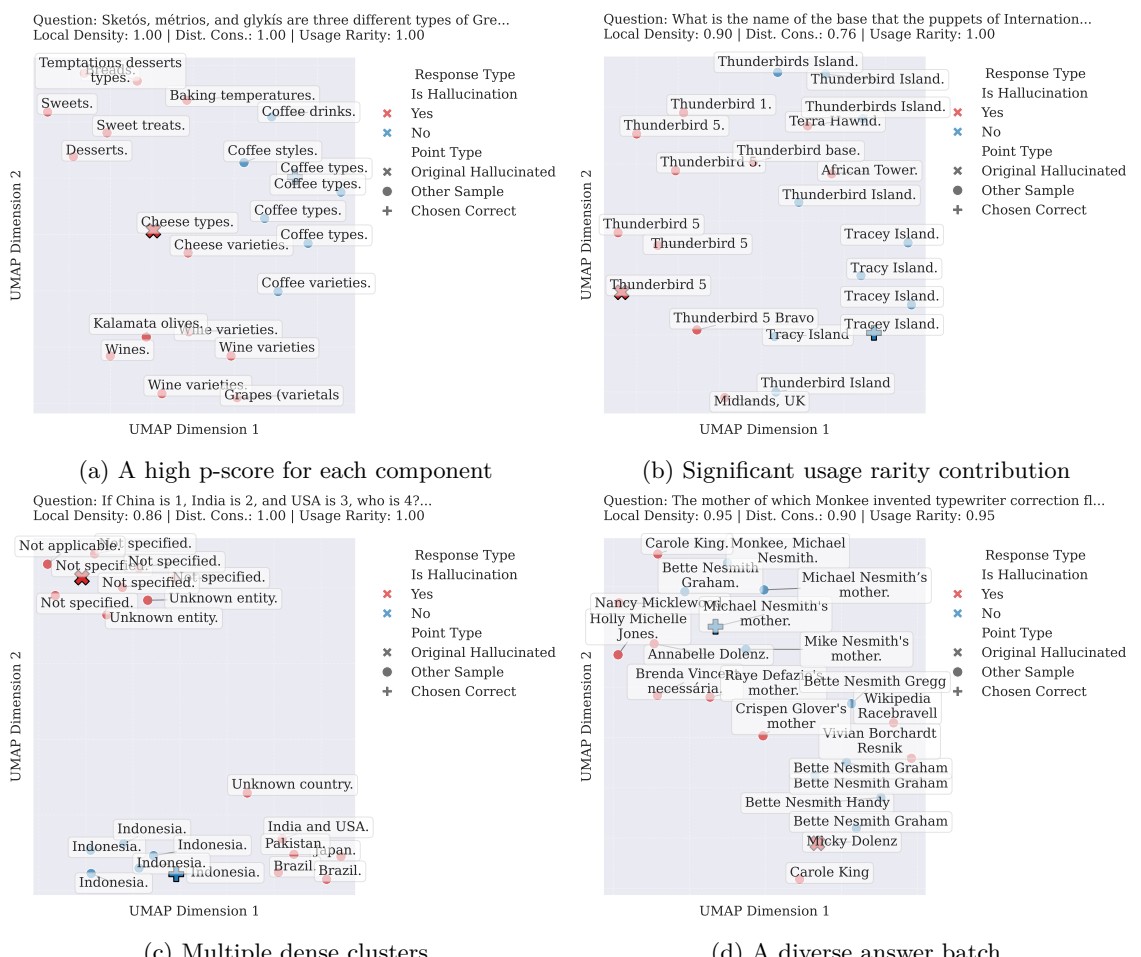

(a) A high p-score for each component

(b) Significant usage rarity contribution

(c) Multiple dense clusters

(d) A diverse answer batch

Figure 4: UMAP plots for cases where we were able to correct hallucinations with our local uncertainty metric, using the TriviaQA dataset and GPT-4o-mini model. In each plot, the red X shows $r_{\text{default}}$, which was determined to be a hallucination by a judge LLM. The other points show the batch of $n = 20$ answers sampled from the same model with a temperature of one. The blue cross shows the answer selected by our framework, which was determined by the same judge LLM to be a non-hallucination.

closer to the correct answer, moving the global consensus further in this direction. Finally, Figure 4(d) shows our framework can be useful in cases of diverse answer batches without dense local clusters. In this example approximately half of the answers are semantically unique, but the use of commonly used archetypes in the 'Michael Nesmith' cluster results in selection of a low suspicion example which proves to be correct.

## 5 Discussion

Our framework has several limitations, which we aim to address in future work. Firstly, it relies on the capability of a single embedding vector to sufficiently capture the semantic content of an answer. As answers get longer and semantically more complex, this assumption may break down. Recent work has investigated this and suggests that the principle of semantic distribution analysis with a single embedding vector holds promise for answers of up to 1000 words (Bhardwaj et al., 2025), but it remains to be proven for complex datasets beyond QA tasks.

Secondly, our method relies on the principle of semantic dispersion in sampled answers predicting hallucination. It is therefore most effective in settings where the model is uncertain and wrong, i.e. there is sufficient

diversity in the sampled answers. As we discovered in our experiments, there exist settings where models tend to be confident and wrong, with no diversity in sampled answers. Additionally, in QA tasks there are often scenarios with multiple, semantically distinct answers to a question which are all correct. White-box methods such as factuality probes (Han et al., 2025) may be more appropriate to address these confidently wrong or 'diversely correct' cases.

Thirdly, we do not characterize how the prompt-conditioned semantic distribution changes as a function of temperature. We would expect the geometric volume metric to increase as batches are sampled at higher temperatures, but the optimal temperature may be model-, dataset-, and task-dependent. We conducted this work under constrained sampling budget; investigating even one other temperature setting would double the compute cost of the experiments. We follow previous work in semantic dispersion-based UQ in selecting $T = 1$ (Farquhar et al., 2024; Kossen et al., 2024) for batch generation.

Finally, our evaluations require hallucination labels for open-ended answers, for which we use ROUGE-L against reference answers for short-form QA (following Li et al. (2025)), and an LLM judge for the medical and long-form settings. Both may introduce label noise: ROUGE-L can penalize correct paraphrases and reward lexical overlap without semantic correctness, and LLM judges introduce their own errors and biases. Label noise affects absolute scores, and where a labelling function shares a bias with an uncertainty method (most notably sensitivity to response length) it can also distort the relative ranking of methods (Santilli et al., 2025). Applying identical labels to all methods removes one source of variation but does not eliminate this risk; we mitigate it by using an LLM judge for the long-form settings, which Santilli et al. (2025) identify as the least length-biased of the common correctness functions, and we note that our qualitative conclusions are consistent across datasets, models, and both labelling schemes. Future work could obtain more reliable labels by marginalizing over multiple judge variants (Ielanskyi et al., 2026).

# 6 Conclusion

We presented a unified geometric framework for quantifying uncertainty in large language models with only black-box access, which explicitly models the support of prompt-conditioned semantic distributions. At the global level, we established *Geometric Volume* as a theoretically grounded upper bound on semantic entropy, offering a robust proxy for hallucination detection. *Geometric Suspicion* leverages the boundaries of this support to identify and correct unreliable responses via a Best-of-N strategy. Our experiments demonstrate strong performance in both standard QA tasks and high-stakes medical domains. These results highlight geometry as a powerful tool for linking global dispersion with local reliability, and open new directions for uncertainty estimation methods that are interpretable, data-efficient, and broadly applicable.

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

## A    Appendix

## B    Theoretical Analyses

**Corollary 1.** *Let a response be represented as a convex combination of archetypes lying in a simplex $\Delta \subset \mathbb{R}^d$ with volume $V > 0$. Then the differential entropy of $\mathbf{x}$, with density $p(\mathbf{x})$ supported on $\Delta$, satisfies:*

$$H(\mathbf{x}) \leq \log V. \tag{20}$$

*Consequently, the volume $V$ provides a geometric upper bound on the differential entropy of a response distribution defined within the simplex. Differential entropy in this context is analogous to the semantic uncertainty of a response set. This bound is computationally efficient to estimate and can be used to detect low-confidence or out-of-distribution inputs without explicitly modeling $p(\mathbf{x})$.*

This result follows directly from Theorem 1, by applying the entropy–volume bound to the simplex associated with the response's archetypal representation.

### B.1    Proof of Theorem 1

*Proof.* Let $u(\mathbf{x}) = \frac{1}{V}$ denote the uniform distribution over $\Delta$, where $V$ is the volume of $\Delta$ under the $(k-1)$-dimensional Hausdorff measure $\mathcal{H}^{k-1}$. The differential entropy of $u$ is:

$$H(u) = - \int_{\Delta} \frac{1}{V} \log\left(\frac{1}{V}\right) d\mathcal{H}^{k-1}(\mathbf{x}) = \log V, \tag{21}$$

since $\int_{\Delta} d\mathcal{H}^{k-1}(\mathbf{x}) = V$.

The Kullback–Leibler (KL) divergence from $p$ to $u$ is:

$$D_{\mathrm{KL}}(p\|u) = \int_{\Delta} p(\mathbf{x}) \log\left(\frac{p(\mathbf{x})}{u(\mathbf{x})}\right) d\mathcal{H}^{k-1}(\mathbf{x}). \tag{22}$$

Substituting $u(\mathbf{x}) = \frac{1}{V}$ yields:

$$D_{\mathrm{KL}}(p\|u) = \int_{\Delta} p(\mathbf{x}) \log p(\mathbf{x}) \, d\mathcal{H}^{k-1}(\mathbf{x}) + \log V. \tag{23}$$

Rearranging terms, we obtain:

$$H(\mathbf{x}) = - \int_{\Delta} p(\mathbf{x}) \log p(\mathbf{x}) \, d\mathcal{H}^{k-1}(\mathbf{x}) = \log V - D_{\mathrm{KL}}(p\|u). \tag{24}$$

By the non-negativity of KL divergence (Gibbs' inequality), $D_{\mathrm{KL}}(p\|u) \geq 0$, with equality if and only if $p = u$ everywhere. Thus:

$$H(\mathbf{x}) \leq \log V, \tag{25}$$

with equality if and only if $\mathbf{x}$ is uniformly distributed over $\Delta$.    □

### B.2    Proof of Proposition  1

*Proof.* Let $d_{\mathrm{int}}$ denote the intrinsic dimension of the archetypal hull $\widehat{S}_q = \mathrm{conv}(\mathbf{Z})$, and let $\mathrm{Vol}(\widehat{S}_q)$ denote its intrinsic $d_{\mathrm{int}}$-volume (Hausdorff measure on the affine span, as in Theorem 1). Under the assumption that the true support $S_q$ is well-approximated by $\widehat{S}_q$, we may treat the semantic distribution $P_q$ as (essentially) supported on $\widehat{S}_q$; in particular, in the idealized case $S_q \subseteq \widehat{S}_q$ this is exact, while any small support mismatch contributes only an additive approximation error that is absorbed by the $\lesssim$ notation.

Applying Theorem 1 to the support set $\widehat{S}_q$ (viewed in its affine span) yields $H(P_q) \leq \log \mathrm{Vol}(\widehat{S}_q)$. This establishes the claimed bound up to the approximation error implicit in $\lesssim$. Finally, since $H_G(\mathbf{X}) = \frac{1}{d_{\mathrm{int}}} \log(\mathrm{Vol}(\widehat{S}_q) + \epsilon)$ differs from $\frac{1}{d_{\mathrm{int}}} \log \mathrm{Vol}(\widehat{S}_q)$ only by a numerical stability term, $H_G(\mathbf{X})$ serves as a theoretically grounded proxy for the worst-case semantic uncertainty consistent with the estimated support.    □

### B.3 Proofs of Lemma 2 and Proposition 2

We first state a standard change-of-variables result for intrinsic volumes under full-rank linear maps, which we will use repeatedly.

**Normalization of intrinsic volume.** Throughout, we use the standard normalization of the $m$-dimensional Hausdorff measure $\mathcal{H}^m$ so that, on any $m$-dimensional affine subspace of $\mathbb{R}^{d'}$, $\mathcal{H}^m$ agrees with $m$-dimensional Lebesgue measure in orthonormal coordinates. Concretely, if $Q \in \mathbb{R}^{d' \times m}$ has orthonormal columns and $b \in \mathbb{R}^{d'}$, then for every Lebesgue-measurable $B \subset \mathbb{R}^m$,

$$\mathcal{H}^m(b + QB) = \mathcal{L}^m(B). \tag{26}$$

**Lemma 4** (Scaling of intrinsic volume under full-rank linear maps)**.** *Let $L : \mathbb{R}^m \to \mathbb{R}^{d'}$ be a linear map with $\mathrm{rank}(L) = m$. Let $S := L(\mathbb{R}^m)$, which is an $m$-dimensional linear subspace of $\mathbb{R}^{d'}$. Then for any Lebesgue-measurable $A \subset \mathbb{R}^m$,*

$$\mathcal{H}^m(LA) \;=\; \sqrt{\det(L^\top L)}\, \mathcal{L}^m(A), \tag{27}$$

*where $\mathcal{L}^m$ is $m$-dimensional Lebesgue measure on $\mathbb{R}^m$, and $\mathcal{H}^m$ is the $m$-dimensional Hausdorff measure restricted to the subspace $S$ (with the normalization in equation 26).*

*Proof.* Since $\mathrm{rank}(L) = m$, $L$ admits a (thin) singular value decomposition

$$L = U\Sigma V^\top, \tag{28}$$

where $U \in \mathbb{R}^{d' \times m}$ has orthonormal columns, $V \in \mathbb{R}^{m \times m}$ is orthogonal, and $\Sigma = \mathrm{diag}(\sigma_1, \ldots, \sigma_m)$ with $\sigma_i > 0$. Let $A \subset \mathbb{R}^m$ be Lebesgue-measurable. Using that $U$ is an isometric embedding of $\mathbb{R}^m$ into $\mathbb{R}^{d'}$ and the normalization equation 26, we have for any measurable $B \subset \mathbb{R}^m$,

$$\mathcal{H}^m(UB) = \mathcal{L}^m(B). \tag{29}$$

Therefore,

$$\mathcal{H}^m(LA) = \mathcal{H}^m(U\Sigma V^\top A) = \mathcal{L}^m(\Sigma V^\top A). \tag{30}$$

Since $V^\top$ is orthogonal on $\mathbb{R}^m$, it preserves Lebesgue measure:

$$\mathcal{L}^m(V^\top A) = \mathcal{L}^m(A). \tag{31}$$

Moreover, $\Sigma$ scales the $i$-th coordinate by $\sigma_i$, hence scales $m$-dimensional Lebesgue volume by $\prod_{i=1}^m \sigma_i$, i.e., for any measurable $B \subset \mathbb{R}^m$,

$$\mathcal{L}^m(\Sigma B) = \Big(\prod_{i=1}^m \sigma_i\Big)\mathcal{L}^m(B). \tag{32}$$

Combining equation 30–equation 32 yields

$$\mathcal{H}^m(LA) = \Big(\prod_{i=1}^m \sigma_i\Big)\mathcal{L}^m(A). \tag{33}$$

Finally, using $L^\top L = V\Sigma^2 V^\top$, we obtain

$$\det(L^\top L) = \det(\Sigma^2) = \prod_{i=1}^m \sigma_i^2, \tag{34}$$

so $\prod_{i=1}^m \sigma_i = \sqrt{\det(L^\top L)}$. Substituting into equation 33 gives equation 27. $\qquad\square$

### B.3.1 Lemma 2 Proof

*Proof of Lemma 2.* Let $m := K - 1$ and define the standard $m$-simplex in $\mathbb{R}^m$ by

$$\Delta_m := \{\mathbf{u} \in \mathbb{R}^m : \mathbf{u} \succeq 0, \ \mathbf{1}^\top \mathbf{u} \leq 1\}. \tag{35}$$

Define the affine map $T : \mathbb{R}^m \to \mathbb{R}^{d'}$ as

$$T(\mathbf{u}) := \mathbf{z}_1 + E\mathbf{u}, \qquad E = [\,\mathbf{z}_2 - \mathbf{z}_1, \ldots, \mathbf{z}_K - \mathbf{z}_1\,]. \tag{36}$$

We first show that

$$T(\Delta_m) = \Delta = \operatorname{conv}(\mathcal{Z}). \tag{37}$$

For any $\mathbf{u} \in \Delta_m$,

$$T(\mathbf{u}) = \mathbf{z}_1 + \sum_{i=1}^{m} u_i(\mathbf{z}_{i+1} - \mathbf{z}_1) \tag{38}$$

$$= \left(1 - \sum_{i=1}^{m} u_i\right)\mathbf{z}_1 + \sum_{i=1}^{m} u_i\,\mathbf{z}_{i+1}, \tag{39}$$

which is a convex combination of $\{\mathbf{z}_1, \ldots, \mathbf{z}_K\}$ since $u_i \geq 0$ and $\sum_i u_i \leq 1$. Hence $T(\Delta_m) \subseteq \operatorname{conv}(\mathcal{Z})$. Conversely, any point in $\operatorname{conv}(\mathcal{Z})$ can be written as $\sum_{k=1}^{K} \alpha_k \mathbf{z}_k$ with $\alpha_k \geq 0$ and $\sum_k \alpha_k = 1$. Setting $u_i = \alpha_{i+1}$ gives $\mathbf{u} \in \Delta_m$ and

$$\sum_{k=1}^{K} \alpha_k \mathbf{z}_k = T(\mathbf{u}), \tag{40}$$

so $\operatorname{conv}(\mathcal{Z}) \subseteq T(\Delta_m)$, proving $T(\Delta_m) = \Delta$.

Because $\mathcal{Z}$ is affinely independent, the columns of $E$ are linearly independent and

$$\operatorname{rank}(E) = m. \tag{41}$$

Translation by $\mathbf{z}_1$ does not change intrinsic $m$-volume, therefore

$$\operatorname{Vol}_m(\Delta) = \mathcal{H}^m(T(\Delta_m)) = \mathcal{H}^m(E\Delta_m). \tag{42}$$

Applying Lemma 4 with $L = E$ yields

$$\operatorname{Vol}_m(\Delta) = \sqrt{\det(E^\top E)}\,\mathcal{L}^m(\Delta_m). \tag{43}$$

Finally, $\Delta_m$ is the simplex with vertices $\{0, e_1, \ldots, e_m\}$, so its Lebesgue volume satisfies

$$\mathcal{L}^m(\Delta_m) = \frac{1}{m!}. \tag{44}$$

Combining the above displays gives

$$\operatorname{Vol}_m(\Delta) = \frac{1}{m!}\sqrt{\det(E^\top E)}, \tag{45}$$

as claimed. $\qquad\square$

### B.3.2 Proof of Proposition 2

*Proof of Proposition 2.* Let $m := K - 1$ and retain the full column-rank edge matrix $E \in \mathbb{R}^{d' \times m}$ from Lemma 2. Define

$$\mathcal{P} := \{E\mathbf{u} : \mathbf{u} \in [0,1]^m\}, \qquad \mathcal{S} := \{E\mathbf{u} : \mathbf{u} \succeq 0, \ \mathbf{1}^\top \mathbf{u} \leq 1\}. \tag{46}$$

*Containment.* If $\mathbf{u} \succeq 0$ and $\mathbf{1}^\top \mathbf{u} \leq 1$, then $0 \leq u_i \leq 1$ for each $i$. Hence $\mathbf{u} \in [0,1]^m$, and therefore $\mathcal{S} \subseteq \mathcal{P}$.

*Volume ratio.* By Lemma 4 applied to $L = E$,

$$\mathrm{Vol}_m(\mathcal{P}) = \mathcal{H}^m(E[0,1]^m) = \sqrt{\det(E^\top E)} \, \mathcal{L}^m([0,1]^m)$$
$$= \sqrt{\det(E^\top E)}, \tag{47}$$

since $\mathcal{L}^m([0,1]^m) = 1$. Similarly, writing $\Delta_m = \{\mathbf{u} \succeq 0 : \mathbf{1}^\top \mathbf{u} \leq 1\}$,

$$\mathrm{Vol}_m(\mathcal{S}) = \mathcal{H}^m(E\Delta_m) = \sqrt{\det(E^\top E)} \, \mathcal{L}^m(\Delta_m)$$
$$= \frac{1}{m!} \sqrt{\det(E^\top E)}, \tag{48}$$

since $\mathcal{L}^m(\Delta_m) = 1/m!$ for the standard simplex. Combining the above displays yields

$$\mathrm{Vol}_m(\mathcal{P}) = m! \, \mathrm{Vol}_m(\mathcal{S}), \tag{49}$$

as required. $\qquad\square$

### B.4 Proofs of Local Metrics

### B.4.1 Proof of Lemma 3

*Proof.* Let $y_j := T(r_j)$ for $j = 1, \ldots, n$ and define

$$p_j := \frac{1 + \sum_{\ell=1}^n \mathbb{I}[y_\ell \geq y_j]}{n+1}. \tag{50}$$

Let $y_{(1)} \geq y_{(2)} \geq \cdots \geq y_{(n)}$ be a non-increasing ordering of the multiset $\{y_1, \ldots, y_n\}$ (breaking ties arbitrarily), and let $p_{(j)}$ denote the $p$-value associated with $y_{(j)}$ via equation 50. For each $j \in \{1, \ldots, n\}$, at least the first $j$ ordered values satisfy $y_{(\ell)} \geq y_{(j)}$, hence

$$p_{(j)} = \frac{1 + \sum_{\ell=1}^n \mathbb{I}[y_\ell \geq y_{(j)}]}{n+1} \geq \frac{1+j}{n+1}. \tag{51}$$

Fix any $t \in [0,1]$ and define $N_t := \sum_{j=1}^n \mathbb{I}[p_j \leq t]$. If $p_{(j)} \leq t$, then by equation 51 we have $(1+j)/(n+1) \leq t$, which implies $j \leq t(n+1) - 1$. Therefore,

$$N_t \leq \max\{0, \lfloor t(n+1) - 1 \rfloor\} \leq \max\{0, t(n+1) - 1\}. \tag{52}$$

Dividing by $n$ and using $t \leq 1$ gives

$$\frac{1}{n} \sum_{j=1}^n \mathbb{I}[p_j \leq t] = \frac{N_t}{n} \leq \max\left\{0, \, t + \frac{t-1}{n}\right\} \leq t. \tag{53}$$

Taking expectations of equation 53 yields

$$\mathbb{E}\left[\frac{1}{n} \sum_{j=1}^n \mathbb{I}[p_j \leq t]\right] \leq t. \tag{54}$$

By exchangeability of $(y_1, \ldots, y_n)$, the events $\{p_j \leq t\}$ have identical probabilities for all $j$, hence

$$\mathbb{P}(p_i \leq t) = \frac{1}{n} \sum_{j=1}^n \mathbb{P}(p_j \leq t) = \mathbb{E}\left[\frac{1}{n} \sum_{j=1}^n \mathbb{I}[p_j \leq t]\right]. \tag{55}$$

Combining equation 54 and equation 55 gives $\mathbb{P}(p_i \leq t) \leq t$ for all $t \in [0,1]$, which is exactly equation 16. $\quad\square$

### B.4.2 Proof of Proposition 3

*Proof.* Writing three component as p-values:

$$p_1 := p_L(r_i), \qquad p_2 := p_D(r_i), \qquad p_3 := p_U(r_i), \tag{56}$$

and note that each $p_j$ is super-uniform, meaning $\mathbb{P}(p_j \leq u) \leq u$ for all $u \in [0,1]$. For any $s \geq 0$, define $u := \exp(-s/2) \in (0,1]$. Using the definition of $S(r_i)$,

$$\mathbb{P}(S(r_i) \geq s) = \mathbb{P}\left(-2\sum_{j=1}^{3} \log p_j \geq s\right) = \mathbb{P}\left(\sum_{j=1}^{3} \log p_j \leq -\frac{s}{2}\right) = \mathbb{P}\left(\prod_{j=1}^{3} p_j \leq u\right). \tag{57}$$

If $\prod_{j=1}^{3} p_j \leq u$, then necessarily $\min_{j \in \{1,2,3\}} p_j \leq u^{1/3}$, since otherwise $p_j > u^{1/3}$ for all $j$ would imply $\prod_{j=1}^{3} p_j > u$. Hence,

$$\mathbb{P}\left(\prod_{j=1}^{3} p_j \leq u\right) \leq \mathbb{P}\left(\min_{j \in \{1,2,3\}} p_j \leq u^{1/3}\right) \leq \sum_{j=1}^{3} \mathbb{P}\left(p_j \leq u^{1/3}\right), \tag{58}$$

where the last inequality is the union bound. By super-uniformity of each $p_j$ and the fact that $u^{1/3} \in (0,1]$, we obtain

$$\sum_{j=1}^{3} \mathbb{P}\left(p_j \leq u^{1/3}\right) \leq \sum_{j=1}^{3} u^{1/3} = 3u^{1/3} = 3\exp\left(-\frac{s}{6}\right). \tag{59}$$

Combining equation 57–equation 59 yields

$$\mathbb{P}(S(r_i) \geq s) \leq 3\exp\left(-\frac{s}{6}\right), \tag{60}$$

which proves equation 17.

We now verify the combined p-value claim. For $t \in [0,1)$, the event $\{p_{\text{comb}}(r_i) \leq t\}$ is equivalent to $\{3\exp(-S(r_i)/6) \leq t\}$, since the alternative case $p_{\text{comb}}(r_i) = 1$ cannot satisfy $1 \leq t < 1$. Therefore,

$$\mathbb{P}(p_{\text{comb}}(r_i) \leq t) = \mathbb{P}\left(3\exp\left(-\frac{S(r_i)}{6}\right) \leq t\right) = \mathbb{P}\left(S(r_i) \geq 6\log\left(\frac{3}{t}\right)\right). \tag{61}$$

Applying equation 17 with $s = 6\log(3/t)$ gives

$$\mathbb{P}(p_{\text{comb}}(r_i) \leq t) \leq 3\exp\left(-\log\left(\frac{3}{t}\right)\right) = 3 \cdot \frac{t}{3} = t, \tag{62}$$

for all $t \in (0,1)$. The case $t = 0$ is trivial, and for $t = 1$ we have $\mathbb{P}(p_{\text{comb}}(r_i) \leq 1) = 1$. Hence $p_{\text{comb}}(r_i)$ is super-uniform on $[0,1]$. Under the exchangeable null, Lemma 3 ensures the required super-uniformity of each component p-value, completing the proof. $\square$

## C Experimental Setup

### C.1 Benchmarks and Baselines

**CLAMBER (External Uncertainty).** To evaluate the detection of external hallucinations arising from ambiguous user prompts, we use the CLAMBER benchmark (Zhang et al., 2024). We use 3,202 queries from the dataset, where the model's uncertainty should reflect the ambiguity of the input rather than a lack of internal knowledge.

**TriviaQA (Internal Uncertainty).** We also evaluate on TriviaQA, a short-form question answering benchmark (Joshi et al., 2017). From the TriviaQA dataset, we construct balanced test sets of 1,000 samples, consisting of 500 hallucinated and 500 non-hallucinated examples.

**ScienceQA Multiple Choice QA**   We also used ScienceQA as a benchmark (Lu et al., 2022), which is a multimodal multiple-choice dataset designed to assess scientific reasoning across natural, social, and language sciences. For our evaluation, we filtered the test set to include only text-only questions and created a balanced benchmark of 200 hallucinating and 200 non-hallucinating examples for each LLM.

**MedicalQA (High-Stakes Domain).**   To assess performance in a critical, high-stakes domain, we use a medical question-answering benchmark subset (500 examples) from MedQA and MedMcQA (Zhang et al., 2023). This dataset tests the model's ability to provide factually correct answers to medical questions, where hallucinations carry significant risk. MedicalQA is a long-form question answering benchmark, unlike TriviaQA, and provides deeper insight into hallucination in real-world applications.

**K-QA (Real-World Medical Scenarios).**   Finally, to evaluate uncertainty in real-world medical scenarios, we leverage 201 real patient questions from the K-QA dataset (Manes et al., 2024), a long-form generation benchmark with expert-written answers reviewed by licensed physicians. These detailed responses serve as high-quality references for assessing the accuracy and reliability of model outputs.

## C.2   Dataset Curation

**External Uncertainty.**   For the CLAMBER benchmark, we evaluate each of the 3,202 queries, where each $q_i$ is annotated with a binary label $y_i \in 0, 1$ indicating whether it is ambiguous. We generate $n = 20$ perturbations $\{q_i^{(j)}\}_{j=1}^n$ for each query by prompting each LLM following previous work (Li et al., 2025), and obtain embeddings $\{x_i^{(j)}\}_{j=1}^n$ via the Alibaba-NLP/gte-Qwen2-1.5B-instruct model.

**Internal Uncertainty.**   To create hallucination detection datasets from traditional QA benchmarks, we convert question–answer pairs $(q_i, a_i^{\text{ref}})$ into labelled examples. For each question $q_i$, we generate a response $r_i = \text{LLM}(q_i)$. We then assign a hallucination label $y_i \in \{0, 1\}$ by comparing $r_i$ to the reference answer $a_i^{\text{ref}}$. In high-stakes domains such as medical QA, this comparison is performed by an LLM-as-a-judge (Gu et al., 2024) (GPT-4o) using expert-annotated references. For datasets like TriviaQA, where answers are typically short (1–3 words), we instead compute ROUGE-L F1 and label $r_i$ as hallucinated if $\text{ROUGE}(r_i, a_i^{\text{ref}}) < 0.3$, following previous work (Li et al., 2025). When initially answering questions, we set the generation temperature to 0. Later, for sampling perturbed responses, we use a temperature of 1.0 to induce diversity. Finally, for each $q_i$, we sample $n = 20$ perturbed responses $\{r_i^{(j)}\}_{j=1}^n$ and embed them via $\{\mathbf{x_i^{(j)}} = \mathcal{E}(r_i^{(j)})\}_{j=1}^n$, using the same embedding pipeline as in the external uncertainty setting.

## C.3   Implementation Details

We set the PCA dimension to 15 and number of archetypes $K = 16$ for each benchmark. The Archetypal Analysis optimization is run for 2000 steps on each batch of sampled responses. For a fair comparison, both our method and the Semantic Volume baseline require an uncertainty threshold $\tau$ to make a binary prediction (hallucination vs. not) (Li et al., 2025). Following prior work, we tune this threshold on a held-out validation split (10% of the data), selecting the $\tau$ that maximizes the F1-score. Local uncertainty experiments are performed with $k = 5$ nearest neighbours for the local density metric.

**Baselines**   For Degree and Eccentricity (Lin et al., 2024), we adapt the code from previous work (`https://github.com/zlin7/UQ-NLG`) and use the entailment variant with the Deberta-Large model (`https://huggingface.co/microsoft/deberta-large` and an eigenvalue threshold of 0.7.

# D   Computational and Memory Complexity

The dominant cost of our framework arises from the archetypal analysis step, which alternates between coefficient and archetype updates. Each update involves matrix multiplications over $n$ sampled responses of reduced dimension $d'$, resulting in a total time complexity of $\mathcal{O}(T n d' K)$, where $T$ is the number of alternating optimization steps and $K$ the number of archetypes. In our setting, $d' < n$ due to dimensionality

reduction, and both $K$ and $T$ are small constants, making the method effectively linear in the number of responses and embedding dimensions.

The memory footprint is dominated by the response embeddings $\mathbf{X} \in \mathbb{R}^{n \times d'}$ and coefficient matrices $\mathbf{A} \in \mathbb{R}^{K \times n}$ and $\mathbf{B} \in \mathbb{R}^{n \times K}$, yielding an overall space complexity of $\mathcal{O}(nd' + nK + Kd')$. Since $K \leq \min(n, d' + 1)$, the approach remains both computationally and memory efficient for large-scale response batches.

## E  Empirical Support for Geometric Suspicion Terms

Our *Geometric Suspicion* score is composed of three separate terms. To support our choice for these terms, we conducted an analysis of the datasets generated for the global uncertainty evaluation.

For each model $M$ and dataset $D$ investigated, we have a set of $N_{D,M}$ questions. For each question, we have a 'default' answer, sampled with a temperature $T = 0$, and a corresponding hallucination label as determined by a judge LLM (as described in Appendix Section C). For each question, we have in addition $n = 20$ answers sampled with temperature $T = 1$, for which we generate hallucination labels as for the default answer. For each batch of answers, we then have the objects derived from our archetypal analysis: the answer embeddings $\mathbf{X}$, the convex decomposition coefficients $\mathbf{A}$, and the archetype embeddings $\mathbf{Z}$.

We first assess the potential to correct an erroneous model response to a non-hallucination by preferential selection of one of the batch responses. Figure 5 plots a histogram of the sampled hallucination rate within our datasets. In the majority of cases, when the model hallucinates in the default answer, a high proportion of the sampled answers are also hallucinations. Approximately half the time, however, at least one answer is a non-hallucination, providing an opportunity for our local metric to be of use.

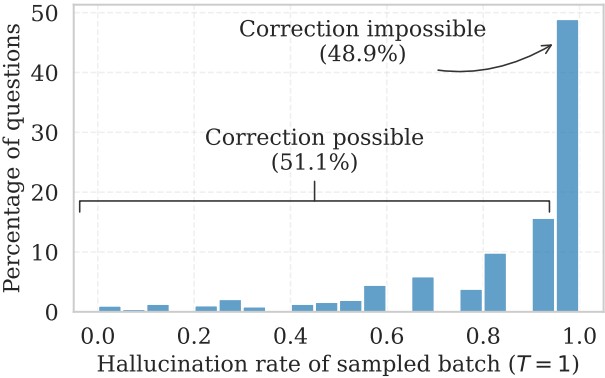

Figure 5: Distribution across all models and datasets of the sampled $T = 1$ hallucination rate when the $T = 0$ answer is hallucinated. Correction is possible in approximately half of cases.

Next, we aimed to find some high level characteristics of hallucinations within our response set. We selected cases where the default response was judged to be a hallucination, but at least one hallucination and non-hallucination was present in the batch of $T = 1$ answers. As introduced in Section 3.4, our local uncertainty metric is based on the guiding principles that hallucinated samples are more likely to be *(i) locally isolated, (ii) far from the dominant semantic consensus*, or *(iii) explained by rare/extremal archetypal directions*.

To test these principles, we compiled several terms of interest and compared them across our datasets. To measure the density of points within the embedding space, we used Local Sparsity with $k = 5$ (as described in Section 3.4) and Voronoi cell volume. The Voronoi cell of an embedding $\mathbf{x}_i$ is the region of space closer to $\mathbf{x}_i$ than to any other point in the batch. A larger cell volume implies the response lies in a sparser region. As direct computation of these volumes in high-dimensional space is intractable, we approximate them via the dual Delaunay triangulation. First, for each batch of embeddings $\mathbf{X} = \{\mathbf{x}_1, \ldots, \mathbf{x}_N\}$, we reduce their dimensionality to a low-dimensional space (typically 3D) using Principal Component Analysis (PCA). We then compute the Delaunay triangulation of these reduced points. The volume of a response's Voronoi cell is approximated by summing the volumes of all Delaunay simplices that have the response's embedding as

a vertex. A larger estimated volume signifies that a response lies in a sparser region of the local embedding space and therefore might be considered more suspicious.

To assess global position within the embedding space, we used the Distance from Consensus term described in Section 3.4 and Distance to Closest Archetype. The latter assesses a response's proximity to the boundaries of the semantic space spanned by the batch. For each response embedding $\mathbf{x}_i$, we compute its Euclidean distance to every archetype embedding $\mathbf{z}_k$, and take the minimum. A smaller distance means the response is closer to the edges of the batch semantic space and indicates greater suspicion. To align this with our other metrics where higher values denote higher suspicion, the final score is the negative of this minimum distance, defined as:

$$D_A(r_i) = - \min_{k \in \{1,\dots,K\}} \|\mathbf{x}_i - \mathbf{z_k}\|_2 \tag{63}$$

To assess the archetypal composition, we used Usage Rarity, as described in Section 3.4, and a new measure we term *Geometric Entropy*, the Shannon entropy (Shannon, 1948) of the archetypal reconstruction coefficients. The $i$-th row of $\mathbf{A}$, denoted $\boldsymbol{\alpha}_i = [A_{i1},\dots,A_{iK}]$, is a probability distribution describing how the pre-processed response embedding $\mathbf{x}_i$ is reconstructed from the set of learned archetypes $\mathbf{Z}$. A response that aligns clearly with a single archetype will have a sparse, low-entropy $\boldsymbol{\alpha}_i$, while a response that lies between multiple archetypes will have a more uniform, high-entropy $\boldsymbol{\alpha}_i$.

The Geometric Entropy for an individual response $r_i$ is defined as:

$$H_L(r_i) = H(\boldsymbol{\alpha}_i) = - \sum_{k=1}^{K} A_{ik} \log A_{ik} \tag{64}$$

Additionally, we split our dataset into subsets based on the sampled hallucination rate. As behaviour such as clustering could be very different depending on whether 1 or 19 of the 20 responses were hallucinations, we split the dataset into *low*, *mid-low*, *mid-high*, and *high* hallucination subsets, with the corresponding rates described in Table 3.

Table 3: Definition of sampled hallucination rate subsets used for granular analysis. The rate, $r$, refers to the fraction of hallucinated responses within a 20-sample set for a given question.

| Subset Name | Sampled Hallucination Rate ($r$) |
|:---:|:---:|
| Low | $0 < r \leq 0.25$ |
| Mid-Low | $0.25 < r \leq 0.50$ |
| Mid-High | $0.50 < r \leq 0.75$ |
| High | $0.75 < r < 1.0$ |

For each term and subset of interest, we then compared the distribution of term values for hallucinations versus non-hallucinations in the sampled answers. In calculating each term, we constructed the sign such that our hypothesis for its predictive value aligned with our local uncertainty framework: a high term value should lead to high suspicion of being a hallucination. Finally, we performed a one-sided Mann-Whitney U Test test to determine whether the term scores for hallucinations were significantly greater than for non-hallucinations. Table 4 shows the results.

For all of our terms of interest, and hallucination rates less than 0.5, our intuitions were proved correct: non-hallucinations lie in denser regions of semantic space, are closer to a global consensus, and generally lie far away from archetypes and the edge of the embedding space. For high hallucination rates (over 0.75) this behaviour is completely reversed. We accept that in these cases it may be extremely hard to find the needle non-hallucination in a haystack of hallucinations. We suggest that in such situations it may be preferential to first use the global uncertainty measure *Geometric Volume* to identify that the batch has high semantic dispersion, and conclude that few if any sampled responses are likely to be reliable.

For the mid-high subset, for all our terms of interest bar 'Distance from Consensus' there is inconclusive evidence that hallucinations have greater values, although Local Density and Usage Rarity have p-values

Table 4: One-sided p-values from the Mann-Whitney U test, testing if metric scores for hallucinations are significantly greater than for non-hallucinations across the defined data subsets (see Table 3). An extremely small p-value (e.g., $< 10^{-100}$) indicates a metric is both statistically significant and directionally correct. Values of 1.000 indicate the metric is significant in the opposite direction (lower scores for hallucinations).

| | Sampled Hallucination Rate Subset | | | |
|---|---|---|---|---|
| **Uncertainty Metric** | **Low** | **Mid-Low** | **Mid-High** | **High** |
| Distance Consensus | $< 10^{-100}$ | $< 10^{-100}$ | 1.000 | 1.000 |
| Local Density | $< 10^{-100}$ | $< 10^{-100}$ | 0.0152 | 1.000 |
| Usage Rarity | $< 10^{-100}$ | $< 10^{-100}$ | 0.0544 | 1.000 |
| Voronoi Volume | $< 10^{-100}$ | $< 10^{-100}$ | 0.0739 | 1.000 |
| Geometric Entropy | $< 10^{-100}$ | $< 10^{-100}$ | 0.2903 | 1.000 |
| Distance Nearest Archetype | $< 10^{-100}$ | $< 10^{-100}$ | 0.2533 | 1.000 |

of 5% or below. This suggests potential to diagnose hallucinations within the batch even at these high hallucination rates, using a combination of multiple terms.

## F  Ablations

### F.1  Global Uncertainty

We report ablations for global uncertainty in Table 5, showing that high $n$ and $d'$ are desirable, with performance fairly constant as the number of archetypes is varied. We choose $K = 16$ in our main results for improved performance in our local metrics.

Table 5: **Sensitivity Analysis on TriviaQA.** We report Mean $\pm$ Std over 3 runs. **(a)** As expected, increasing the sample size leads to improved performance. **(b)** The method requires $d' \geq 5$ to capture semantic structure; the poor performance at $d' = 2$ validates the necessity of high-dimensional volume over simple 2D area. **(c)** Geometric Volume is robust to the number of archetypes $K$, remaining stable even with a minimal simplex ($K = 4$).

| (a) Sample Size ($n$) | | | | (b) PCA Dim ($d'$) | | | | (c) Archetypes ($K$) | | |
|---|---|---|---|---|---|---|---|---|---|---|
| $n$ | AUROC | F1 | | $d'$ | AUROC | F1 | | $K$ | AUROC | F1 |
| 5 | $0.705 \pm 0.010$ | $0.652 \pm 0.017$ | | 2 | $0.692 \pm 0.014$ | $0.666 \pm 0.002$ | | 4 | $\mathbf{0.753} \pm 0.001$ | $0.706 \pm 0.001$ |
| 10 | $0.731 \pm 0.004$ | $0.674 \pm 0.012$ | | 5 | $0.746 \pm 0.005$ | $0.701 \pm 0.007$ | | 8 | $\mathbf{0.753} \pm 0.002$ | $0.704 \pm 0.004$ |
| 15 | $0.744 \pm 0.003$ | $0.682 \pm 0.014$ | | 10 | $0.752 \pm 0.003$ | $\mathbf{0.705} \pm 0.004$ | | 12 | $\mathbf{0.753} \pm 0.002$ | $\mathbf{0.707} \pm 0.002$ |
| 20 | $\mathbf{0.752} \pm 0.002$ | $\mathbf{0.701} \pm 0.004$ | | 15 | $\mathbf{0.753} \pm 0.002$ | $\mathbf{0.705} \pm 0.002$ | | 16 | $\mathbf{0.753} \pm 0.002$ | $0.705 \pm 0.004$ |

### F.2  Local Uncertainty

We probe robustness of our local uncertainty estimator along three axes: number of sampled answers, PCA dimension, and number of archetypes.

First, Table 6 varies the number of sampled answers $n$. We maintain our fixed answer pool of 20 answers from main experiments: subsets of size $n$ are sampled uniformly without replacement from each question's fixed answer pool. When $n < n_{\text{all}}$ we evaluate $R = 30$ repeated subsets per question. We fit PCA on each subset, so the effective dimensionality obeys $d' \leq n - 1$. We scale the archetype count as $K = \text{round}(\rho n)$ with $\rho = K_{\text{base}}/n_{\text{base}}$ (16 / 20 for our chosen settings), then clip K to satisfy the assumptions of our theorem: $K \leq n$ and $K \leq d' + 1$, ensuring a non-degenerate, identifiable simplex in $d'$ dimensions. As expected, the hallucination correction rate increases with higher $n$, as the sampled set becomes more likely to contain one or more correct answers.

Table 6: Ablation over the number of sampled answers $n$ per question. For each subset we refit PCA on the subset only. We report mean±std AUROC/AUARC across runs and the mean $\Delta H$.

| $n$ | AUROC | AUARC | $\Delta H$ |
|---|---|---|---|
| 5 | $0.540 \pm 0.019$ | $0.576 \pm 0.017$ | 0.012 |
| 10 | $0.550 \pm 0.012$ | $0.593 \pm 0.015$ | 0.031 |
| 15 | $0.547 \pm 0.008$ | $0.587 \pm 0.019$ | 0.039 |
| 20 | $0.556 \pm 0.007$ | $0.585 \pm 0.019$ | 0.068 |

Table 7: Ablation over PCA dimension $d'$ at fixed $n = 20$. We fit a single PCA basis on all 20 answers and compare by truncation. We fix $K = 3$ so results remain feasible down to $d' = 2$.

| $d'$ | AUROC | AUARC | $\Delta H$ |
|---|---|---|---|
| 2 | $0.518 \pm 0.005$ | $0.564 \pm 0.017$ | $-0.021$ |
| 3 | $0.521 \pm 0.010$ | $0.567 \pm 0.015$ | $-0.022$ |
| 5 | $0.535 \pm 0.003$ | $0.575 \pm 0.017$ | 0.044 |
| 10 | $0.551 \pm 0.008$ | $0.581 \pm 0.019$ | 0.032 |
| 15 | $0.554 \pm 0.003$ | $0.585 \pm 0.018$ | 0.062 |

Second, Tables 7 and 8 vary the PCA dimension $d'$ at fixed $n = 20$: we show a fixed-$K$ view ($K = 3$) and a capacity-matched view ($K = \min(d' + 1, K_{\text{base}}, n)$) that respects the simplex capacity of a $d'$-dimensional space. In both cases, increasing $d'$ up to our chosen setting improves the hallucination detection performance. We suggest this is due to low PCA dimensions being unable to capture sufficient geometric details of the semantic answer space.

Finally, Table 9 sweeps the number of archetypes $K$ at fixed $d'$ (default $d' = 15$) while enforcing the same feasibility limits. Interestingly our local metric is able to perform well even with very low numbers of archetypes; we suggest this is due to two of the three component terms using the raw answer embeddings, such that altering the 'usage rarity' term does not drastically affect results.

**Statistics.** For each setting we compute per–question metrics (AUROC, AUARC, and $\Delta H$) and then aggregate. In Table 6 (varying $n$), each question is evaluated on $R = 30$ i.i.d. subsets of size $n$ sampled uniformly without replacement from the fixed 20 answers; we refit PCA on each subset, compute the local score, and obtain the metric values per subset, then average across the $R$ subsets to get a single per–question estimate. Within a run, we then average across questions that pass the filter. Finally, the tables report mean±std across 3 independent runs. In Tables 7, 8 and 9 there is no within–question resampling (one evaluation per question); we average across questions within a run and report mean±std across runs.

Table 8: Ablation over PCA dimension $d'$ at fixed $n = 20$ with capacity-matched archetypes: $K = \min(d' + 1, K_{\text{base}}, n)$. This reflects the geometric limit that at most $d' + 1$ archetypes can be supported in a $d'$-dimensional space.

| $d'$ | AUROC | AUARC | $\Delta H$ |
|---|---|---|---|
| 2 | $0.519 \pm 0.003$ | $0.565 \pm 0.019$ | $-0.018$ |
| 3 | $0.520 \pm 0.007$ | $0.568 \pm 0.016$ | 0.016 |
| 5 | $0.534 \pm 0.003$ | $0.573 \pm 0.020$ | 0.025 |
| 10 | $0.552 \pm 0.005$ | $0.583 \pm 0.018$ | 0.052 |
| 15 | $0.559 \pm 0.004$ | $0.586 \pm 0.021$ | 0.068 |

Table 9: Ablation over the number of archetypes $K$ at fixed $n = 20$ and fixed $d'$ (default $d' = 15$). We enforce feasibility $K \leq n$ and $K \leq d' + 1$.

| $K$ | AUROC | AUARC | $\Delta H$ |
|---|---|---|---|
| 5 | $0.555 \pm 0.007$ | $0.584 \pm 0.020$ | 0.046 |
| 8 | $0.556 \pm 0.003$ | $0.585 \pm 0.019$ | 0.063 |
| 12 | $0.556 \pm 0.003$ | $0.586 \pm 0.020$ | 0.054 |
| 16 | $0.556 \pm 0.003$ | $0.585 \pm 0.020$ | 0.064 |

Table 10: **Robustness of the geometric uncertainty method to the sentence-embedding model.** Mean $\pm$ Std over three runs on MedicalQA and K-QA, with GPT-4o Mini and Llama3.1-8b generators. We report the global Geometric Volume AUROC, the local Geometric Suspicion AUARC, and $\Delta H$ (hallucination-rate reduction under best-of-$N$ rejection).

| Dataset / Model | Embedding model | Global AUROC | Local AUARC | Local $\Delta H$ |
|---|---|---|---|---|
| MedicalQA
GPT-4o Mini | gte-Qwen2-1.5B | $0.596 \pm 0.005$ | $0.586 \pm 0.020$ | $0.098 \pm 0.019$ |
| | EmbeddingGemma-300M | $0.615 \pm 0.014$ | $0.582 \pm 0.018$ | $0.095 \pm 0.017$ |
| | Qwen3-Embed-0.6B | $0.567 \pm 0.014$ | $0.585 \pm 0.023$ | $0.083 \pm 0.014$ |
| | all-MiniLM-L6-v2 | $0.597 \pm 0.012$ | $0.595 \pm 0.019$ | $0.113 \pm 0.019$ |
| MedicalQA
Llama3.1-8b | gte-Qwen2-1.5B | $0.681 \pm 0.013$ | $0.485 \pm 0.013$ | $0.050 \pm 0.020$ |
| | EmbeddingGemma-300M | $0.665 \pm 0.017$ | $0.471 \pm 0.010$ | $0.048 \pm 0.016$ |
| | Qwen3-Embed-0.6B | $0.654 \pm 0.017$ | $0.481 \pm 0.007$ | $0.069 \pm 0.021$ |
| | all-MiniLM-L6-v2 | $0.679 \pm 0.008$ | $0.486 \pm 0.012$ | $0.057 \pm 0.013$ |
| K-QA
GPT-4o Mini | gte-Qwen2-1.5B | $0.671 \pm 0.008$ | $0.641 \pm 0.004$ | $0.158 \pm 0.012$ |
| | EmbeddingGemma-300M | $0.707 \pm 0.016$ | $0.634 \pm 0.006$ | $0.148 \pm 0.019$ |
| | Qwen3-Embed-0.6B | $0.667 \pm 0.013$ | $0.639 \pm 0.005$ | $0.141 \pm 0.042$ |
| | all-MiniLM-L6-v2 | $0.665 \pm 0.005$ | $0.639 \pm 0.008$ | $0.158 \pm 0.018$ |
| K-QA
Llama3.1-8b | gte-Qwen2-1.5B | $0.704 \pm 0.018$ | $0.468 \pm 0.012$ | $0.151 \pm 0.040$ |
| | EmbeddingGemma-300M | $0.659 \pm 0.032$ | $0.474 \pm 0.017$ | $0.172 \pm 0.043$ |
| | Qwen3-Embed-0.6B | $0.651 \pm 0.005$ | $0.471 \pm 0.006$ | $0.161 \pm 0.021$ |
| | all-MiniLM-L6-v2 | $0.695 \pm 0.016$ | $0.471 \pm 0.009$ | $0.161 \pm 0.030$ |

### F.3 Embedding Model

Our method is heavily reliant on the use of a sentence embedding model to produce a semantic representation of each sampled answer. To test the robustness of the method to the choice of this model, we rerun the pipeline while swapping only the embedding model and holding all other components fixed: PCA dimension $d' = 15$, $K = 16$ archetypes, and the same $(L, D, U)$ Geometric Suspicion combination.

We compare four models: our default gte-Qwen2-1.5B[1], EmbeddingGemma-300M[2], Qwen3-Embedding-0.6B[3], and the 22M-parameter all-MiniLM-L6-v2[4]. We evaluate on MedicalQA and K-QA with two generator models each (GPT-4o Mini and Llama3.1-8b), reporting the global Geometric Volume AUROC, the local Geometric Suspicion AUARC, and $\Delta H$ over three runs.

As shown in Table 10 and Figure 6, performance is essentially unchanged across embedders: within every dataset/generator cell the three metrics are largely stable, and the small all-MiniLM-L6-v2 matches the 1.5B baseline throughout. Variation is driven almost entirely by task difficulty rather than by the embedder, and no single embedding model is consistently best. We conclude that the method is robust to the embedding backbone and remains effective with lightweight, inexpensive embedders.

---

[1] https://huggingface.co/Alibaba-NLP/gte-Qwen2-1.5B-instruct
[2] https://huggingface.co/google/embeddinggemma-300m
[3] https://huggingface.co/Qwen/Qwen3-Embedding-0.6B
[4] https://huggingface.co/sentence-transformers/all-MiniLM-L6-v2

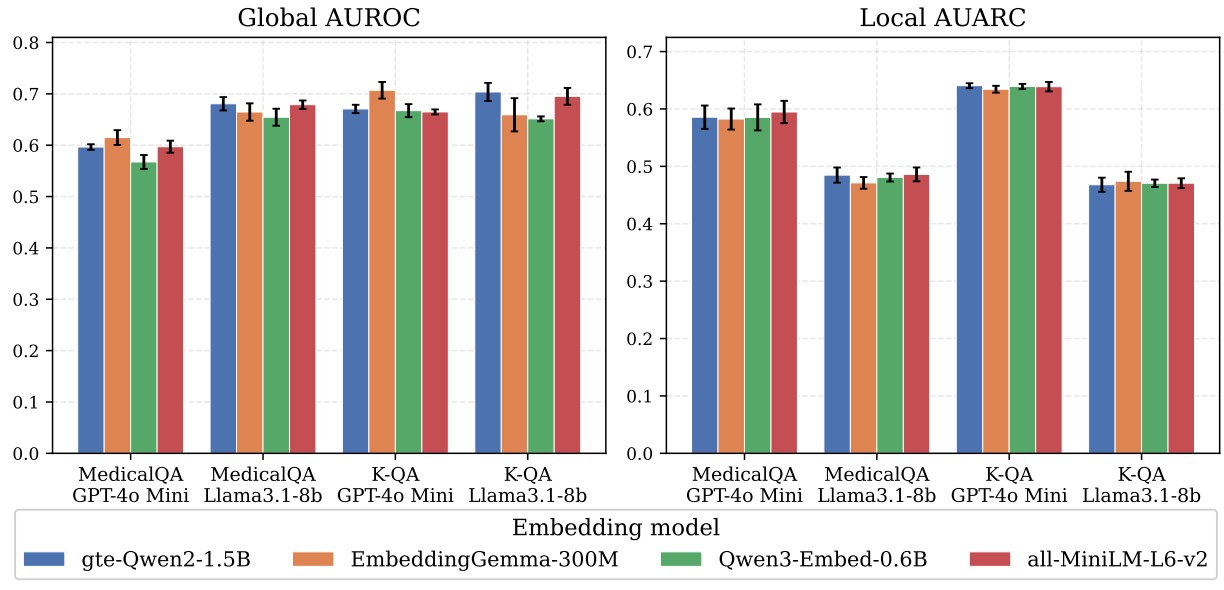

Figure 6: **Robustness to the choice of embedding model.** Global detection AUROC (left) and local AUARC (right) across four sentence-embedding models, grouped by dataset/generator cell. Within every group the metrics are stable across embedders. Error bars denote ±1 standard deviation over 3 runs.

