# OpenReview forum: "Geometric Uncertainty for Detecting and Correcting Hallucinations in LLMs"
_TMLR — Under review for TMLR_

### Review · Reviewer_ZKc2 · 2026-04-05

**Summary Of Contributions:**

The submission views the problem of detecting hallucination from a geometric perspective, which is a novel perspective. Specifically, they use a batch of responses for a given query to form a convex hull in the representation space; they then use the volume of the convex hull as the metric to measure hallucination; they finally use a set of local uncertainty metrics to find out the most reliable response.

**Additional Comments:**

N/A

**Audience:**

Yes

**Audience Explanation:**

The audience who is interested in detecting and mitigating hallucination will be interested in this paper.

**Claims And Evidence:**

Yes

**Claims Explanation:**

The submission provides comprehensive theoretical evidence and extensive empirical evidence to support their claims.

**Requested Changes:**

1. The introduction section could be polished in the sense that the contributions are around global and local uncertainty; however, the definitions for global and local uncertainty are not quite clear in the introduction section. For example, when I read the fourth paragraph in the introduction section, I thought the “batch” refers to a batch of queries, but then I figured out it refers to a batch of responses for a given query. It would be appreciated if the authors could polish the clarification on global and local uncertainty in the introduction section.

2. The selected baselines are few in the sense that some representative hallucination detection methods like [1] are not involved.

3. In addition to AUC and F1, it seems Accuracy itself is a common metric (see [1]). I wonder why it is not used as an evaluation metric in the submission.

[1] Hou, Bairu, et al. "A probabilistic framework for llm hallucination detection via belief tree propagation." Proceedings of the 2025 Conference of the Nations of the Americas Chapter of the Association for Computational Linguistics: Human Language Technologies (Volume 1: Long Papers). 2025.

---

> ### Author Response · Authors · 2026-07-05
> **Author response to ZKc2**
>
> We thank the reviewer for the positive assessment and for concrete suggestions to improve the paper's clarity.
>
> ## Requested changes
>
> **1. Clarifying global vs. local uncertainty in the introduction**
>
> > ...when I read the fourth paragraph in the introduction section, I thought the “batch” refers to a batch of queries, but then I figured out it refers to a batch of responses for a given query. It would be appreciated if the authors could polish the clarification on global and local uncertainty in the introduction section.
>
> We agree the introduction was ambiguous about what a "batch" refers to, and have adjusted the beginning of paragraph four in the introduction to read as follows:
>
> > Within sampling-based methods, where a batch of responses is generated per prompt, we additionally distinguish between two types of uncertainty estimate. A *global* estimate summarizes the uncertainty of the batch, whereas a *local* estimate scores each individual response relative to the others in the same batch.
>
> **2. Additional baselines**
>
> > The selected baselines are few in the sense that some representative hallucination detection methods like [1] are not involved.
>
> We appreciate the pointer to Hou et al. (2025, belief-tree propagation). Our baseline set was chosen to compare against methods that operate in the *same setting* as ours — black-box and sampling-based, requiring only sampled responses rather than model weights or gradients — and to span the principal families of such methods: semantic-clustering (Semantic Entropy, Farquhar et al., 2024), geometric/dispersion-based (Semantic Volume, Li et al., 2025 — our closest competitor), self-evaluation (P(true), Kadavath et al., 2022), and graph-based local measures (Degree and Eccentricity, Lin et al., 2024). This gives directly comparable representatives across families at both the global and local level.
>
> We believe BTProp falls outside this comparable setting, for three reasons. (i) It targets a different problem: it verifies a single declarative statement by recursively decomposing it into logically related sub-claims and running probabilistic inference over a belief tree, whereas our method quantifies dispersion over a batch of sampled answers to a query. (ii) It relies on the model self-scoring its decomposed sub-claims — a self-evaluation signal that is already represented in our comparison by the P(true) baseline. (iii) It requires decomposable declarative claims, whereas most of our benchmarks are short-form QA whose single-entity answers do not decompose into meaningful belief trees.
>
> We welcome further discussion if the reviewer feels we have misunderstood an important reason why our method requires direct comparison to BTProp.
>
> **3. Adding Accuracy as a metric.**
>
> > In addition to AUC and F1, it seems Accuracy itself is a common metric (see [1]). I wonder why it is not used as an evaluation metric in the submission.
>
> We chose AUROC and F1 as our primary metrics for two reasons. First, the central output of a UQ method is a continuous score that ranks responses by reliability; AUROC measures this ranking directly and is independent of any threshold. Second, for a thresholded summary we prefer F1 to Accuracy because some of our detection tasks are class-imbalanced: Accuracy can be inflated by the majority class, whereas F1 emphasizes the positive (hallucination) class and is more informative under imbalance. Accuracy at our tuned threshold is directly derivable from the same predictions, so we could report it alongside F1; we omitted it mainly to avoid further crowding tables that are already quite dense. If the reviewer feels strongly that an accuracy table is required, we are open to including one, perhaps in the Appendix.

---

> > ### Comment · Reviewer_ZKc2 · 2026-07-06
> >
> > Thank the authors for the responses, which have addressed my concerns.

---

### Review · Reviewer_1iGV · 2026-06-25

**Summary Of Contributions:**

The paper proposes a black-box, sampling-based framework for LLM uncertainty that operates in semantic embedding space. For each prompt, it samples multiple responses, embeds them, fits archetypal analysis, and uses the convex hull of the learned archetypes to define a prompt-level uncertainty score, Geometric Volume. It also introduces a response-level score, Geometric Suspicion, which combines local sparsity, distance from consensus, and archetype-usage rarity to rank answers for best-of-(N) selection. The paper further gives theoretical results connecting log-volume to an upper bound on differential entropy and evaluates the method on CLAMBER, TriviaQA, ScienceQA, MedicalQA, and K-QA across four LLMs, with additional ablations in the appendix.

Key strengths are the unified treatment of global and local uncertainty, the black-box applicability, the intuitive geometric interpretation, and a reasonably broad empirical study that includes high-stakes medical settings and ablations over sample count, PCA dimension, and archetype count.

Key weaknesses are that the theory depends on strong assumptions about embedding-space support and how well the archetypal hull approximates it; several labels rely on ROUGE or an LLM judge; threshold tuning needs labeled validation data; and the local correction method only works when the sampled batch already contains a correct answer, which the paper itself identifies as a substantial limitation.

**Additional Comments:**

Overall, I found the paper thoughtful, technically interesting, and reasonably well executed. The geometric perspective is more unified than many prior black-box UQ approaches, and the medical-domain experiments make the contribution practically relevant. My main reservation is that the paper’s strongest evidence supports claims about uncertainty in embedding-space response geometry, whereas some of the presentation occasionally edges toward broader claims about hallucination detection in general. With clearer positioning and a stronger treatment of evaluation assumptions and failure modes, I would view the paper more favorably.

**Audience:**

Yes

**Audience Explanation:**

I think at least part of the TMLR audience would be interested in this paper. It addresses a central topic in current ML research—uncertainty quantification and hallucination mitigation for LLMs—while contributing a new geometric viewpoint that is both black-box and practically motivated.

**Broader Impact Concerns:**

I do not have a major ethical objection to publication.

**Claims And Evidence:**

Yes

**Claims Explanation:**

The main empirical claims are generally supported by the presented evidence. The results tables show that the proposed global metric is usually competitive with or better than the selected baselines, often with especially strong results on the medical benchmarks, and the appendix includes useful ablations that examine sensitivity to sample size, PCA dimension, and the number of archetypes. The theory sections are also clearly written and do provide mathematical justification for the entropy upper-bound interpretation and the rank-based aggregation used in the local score.

That said, some claims should be phrased more carefully. The theoretical argument depends on the semantic distribution being well approximated by the archetypal hull in embedding space, which is a meaningful modeling assumption rather than a direct factual guarantee. Empirical gains are not uniform across all models and datasets, and the labeling pipeline depends on ROUGE-L for some tasks and GPT-4o as judge for others, which introduces some uncertainty into the evaluation itself. So I would mark the answer as Yes, but with important caveats about assumptions, labeling, and generalization.

**Requested Changes:**

**Critical to my recommendation**

1. The paper should tighten its claim language around the theory. The entropy guarantee is about distributions supported on, or well approximated by, the estimated geometric support in embedding space; this should be emphasized as an assumption, not presented too directly as a guarantee about hallucination or real-world factuality.

2. The evaluation methodology needs a stronger discussion of label quality. In particular, the dependence on ROUGE-L for short-answer factuality and GPT-4o as judge for medical/long-form settings should be justified more carefully, and the likely effect of label noise on conclusions should be discussed explicitly.

3. The practical role of threshold tuning should be clarified. The global detector requires selecting a threshold on a labeled validation split, so the method is not fully calibration-free in deployment; this supervision requirement should be made more explicit in the framing.

4. The “confidently wrong” failure mode should be elevated in the main paper. The appendix and discussion make clear that local correction can fail badly when the sampled batch contains few or no correct candidates, and this limitation is central to the claimed correction use case.

**Would strengthen the work but are not strictly critical**

5. Add either a stronger justification for the baseline set or a somewhat broader comparison to recent black-box uncertainty or self-consistency style methods.

6. Report statistical significance tests, or at least stronger pairwise uncertainty analysis for the headline comparisons in the main tables.

---

> ### Author Response · Authors · 2026-07-05
> **Author response to 1iGV (1/2)**
>
> We thank the reviewer for the detailed analysis and concrete suggestions for how to improve the paper.
>
>  ## Requested changes
>
>
> **1. Tightening the theoretical claims**
> > The paper should tighten its claim language around the theory. The entropy guarantee is about distributions supported on, or well approximated by, the estimated geometric support in embedding space; this should be emphasized as an assumption, not presented too directly as a guarantee about hallucination or real-world factuality.
>
> We agree that the entropy result bounds the differential entropy of the *embedded response distribution* under the assumption that it is supported on — or well approximated by — the estimated archetypal hull, something we explicitly note in Proposition 1. To underline this point further, we add the following line after Proposition 1:
>
> > *We stress however that Proposition 1 is a statement about the entropy of the embedded response distribution: the log-volume upper-bounds the semantic entropy only to the extent that the archetypal hull faithfully approximates the true support.*
>
>
> We also agree that semantic dispersion is merely a proxy for hallucination probability, and that we do not offer theoretical guarantees relating to factuality or correctness. To make this more explicit, we now close the Related Work section with the following statement:
>
> > We note however that our method uses the semantic dispersion of the sampled responses as a **proxy** for hallucination risk, rather than a guarantee of factual incorrectness.
>
> We elaborate again on this point in the second paragraph of the discussion section.
>
> **2. Discussion of label quality**
> > The evaluation methodology needs a stronger discussion of label quality. In particular, the dependence on ROUGE-L for short-answer factuality and GPT-4o as judge for medical/long-form settings should be justified more carefully, and the likely effect of label noise on conclusions should be discussed explicitly.
>
> We agree, and have added a paragraph on the labelling pipeline and its limitations to the discussion section. To justify our choices: we use ROUGE-L for short-form QA following prior work (Li et al., 2025), which also enables direct comparison with Semantic Volume, and an LLM judge for the medical and long-form settings, where surface-form matching penalizes correct paraphrases.
>
> On the effect of label noise: applying identical labels to every method removes one source of variation between methods, but we do not claim this makes the rankings immune to label noise. As Santilli et al. (2025) show, when a correctness function and an uncertainty method share a bias — most notably sensitivity to response length — label noise can distort the relative ranking of UQ methods, not just absolute scores. Two considerations mitigate this risk in our setting. First, for the long-form settings we use an LLM judge, which Santilli et al. identify as the least length-biased of the common correctness functions, minimizing the dominant known mechanism of ranking distortion. Second, our qualitative conclusions are consistent across five datasets, four base models, and both labelling schemes, making it unlikely that they are artefacts of a particular correctness function. We nonetheless accept that the labels remain approximate: Ielanskyi et al. (2026) show that marginalizing over multiple LLM-judge variants further reduces evaluation bias, and we now note this explicitly as future work. The revised discussion paragraph reads:
>
> > Finally, our evaluations require hallucination labels for open-ended answers, for which we use ROUGE-L against reference answers for short-form QA (following Li et al., 2025), and an LLM judge for the medical and long-form settings. Both may introduce label noise: ROUGE-L can penalize correct paraphrases and reward lexical overlap without semantic correctness, and LLM judges introduce their own errors and biases. Label noise affects absolute scores, and where a labelling function shares a bias with an uncertainty method — most notably sensitivity to response length — it can also distort the relative ranking of methods (Santilli et al., 2025). Applying identical labels to all methods removes one source of variation but does not eliminate this risk; we mitigate it by using an LLM judge for the long-form settings, which Santilli et al. (2025) identify as the least length-biased of the common correctness functions, and we note that our qualitative conclusions are consistent across datasets, models, and both labelling schemes. Future work could obtain more reliable labels by marginalizing over multiple judge variants (Ielanskyi et al., 2026).

---

> > ### Author Response · Authors · 2026-07-05
> > **Author response to 1iGV (2/2)**
> >
> > **3. Threshold tuning and calibration**
> > > The practical role of threshold tuning should be clarified. The global detector requires selecting a threshold on a labeled validation split, so the method is not fully calibration-free in deployment; this supervision requirement should be made more explicit in the framing.
> >
> > We agree. The global detector is unsupervised in producing the uncertainty *score*, but selecting the operating threshold τ requires a small labelled split, so it is not fully calibration-free in deployment. We have elevated the Appendix implementation detail text so section 4.2 now reads as follows:
> >
> > > *"The threshold is selected on a small labelled validation split (a 10\% subset) by maximising F1. We emphasize that while the Geometric Volume score requires no supervision, this thresholding step does, hence the detector is not calibration-free in deployment."*
> >
> >
> > **4. Elevating the "confidently wrong" failure mode**
> > > The "confidently wrong" failure mode should be elevated in the main paper. The appendix and discussion make clear that local correction can fail badly when the sampled batch contains few or no correct candidates, and this limitation is central to the claimed correction use case.
> >
> > We agree this limitation is central to the correction use case and have promoted it from the appendix into the main text, immediately following the local uncertainty main results.
> >
> >
> > **5. Justification for the baseline set**
> > > Add either a stronger justification for the baseline set or a somewhat broader comparison to recent black-box uncertainty or self-consistency style methods.
> >
> > We have aimed to make our selection criteria explicit. Our baselines were chosen to compare against methods in the *same setting* as ours — black-box and sampling-based, requiring only sampled responses rather than weights or gradients — and to span the principal families of such methods: semantic-clustering (Semantic Entropy, Farquhar et al., 2024), geometric/dispersion-based (Semantic Volume, Li et al., 2025 — our closest competitor), self-evaluation (P(true), Kadavath et al., 2022), and graph-based local measures (Degree and Eccentricity, Lin et al., 2024). We note self-consistency is closely related to the agreement signal captured by these semantic baselines. We have updated section 4.1 with a line that reads as follows:
> >
> > > *"We selected baselines that operate in the same setting as our method (black-box access only, with no model weights or gradients) and that together span differing families of sampling-based uncertainty estimation: semantic clustering (Semantic Entropy), geometric dispersion-based (Semantic Volume), self-evaluation (P(true)), and graph-based local measures (Degree and Eccentricity)."*
> >
> > **6. Statistical significance**
> > > Report statistical significance tests, or at least stronger pairwise uncertainty analysis for the headline comparisons in the main tables.
> >
> > We report mean ± standard deviation over three runs for every entry, which we hope enables pairwise assessment between methods. We emphasize that no single method dominates across all model/dataset combinations, and much of our contribution is the unified framework and geometric interpretation rather than a headline SOTA margin. We suggest that the claims we make therefore do not rest on formal pairwise significance testing.
> >
> > We hope these revisions address the reviewer's concerns, and we would be grateful to know whether they resolve the points raised or whether further changes would elevate the paper. We are open to further discussion on all the above topics in case we have misunderstood a request.

---

### Review · Reviewer_NTGV · 2026-07-01

**Summary Of Contributions:**

This paper introduces a geometric framework to quantify uncertainty in LLMs, consisting of Geometric Volume and Geometric Suspicion. It utilizes archetypal analysis on a batch of sampled response embeddings to construct a geometric support representing the semantic distribution of the answers.
The proposed method is sound and does not rely on access to the model weights of LLMs. However, it is limited by its huge running cost and narrow scope.

**Audience:**

Yes

**Audience Explanation:**

This is the biggest strength of the paper. LLMs are the hottest topic in machine learning and are the most used type of ML models nowadays. A method with potential to detect and minimize hallucinations in LLMs is incredibly helpful for both academic researchers and industrial usage. That said, this method requires repeated sampling, which means roughly an order of magnitude increase in inference cost. For now, this heavily limits its usefulness beyond research purposes. Researchers working in safety and reliability of ML models will appreciate this work.

**Claims And Evidence:**

Yes

**Claims Explanation:**

The theoretical justification is sound, and the experimental results support the claims of the paper. However, the scope of application of this framework to LLMs can be narrow due to the reliance on embedding models to do the heavylifting (the extraction of meanings into a vector embedding). Thus, results are only shown on more simple QA tasks. The theoretical justification is sound but also narrow in scope due to the assumption that uncertainty=hallucinations. This means the method has to rely on the knowledge of the model itself to check for hallucinations. Confident models would pass through this detection system with no way to fix, as it does not make use of any other external sources. These limitations are mentioned in the discussion section of the paper.

**Requested Changes:**

Given how much of the heavy lifting the embedding model is doing in this pipeline, testing and comparing different embedding models could make the result stronger and more robust. This will strengthen the work in my view

---

> ### Author Response · Authors · 2026-07-05
> **Author response to NTGV**
>
> We thank the reviewer for recognizing the soundness of the method and its relevance to LLM safety and reliability, and for the useful observation that the embedding model does substantial work in our pipeline.
>
> ## Requested changes
>
> > Given how much of the heavy lifting the embedding model is doing in this pipeline, testing and comparing different embedding models could make the result stronger and more robust. This will strengthen the work in my view
>
> We agree, and have added an embedding model ablation as a new subsection in the Appendix (F.3). We re-run the full pipeline while swapping only the embedding model and holding everything else fixed (d'=15, K=16, the same (L, D, U) combination), comparing four encoders: our default gte-Qwen2-1.5B, EmbeddingGemma-300M, Qwen3-Embedding-0.6B, and the 22M-parameter all-MiniLM-L6-v2. We evaluate on MedicalQA and K-QA with two generator models each (GPT-4o Mini and Llama3.1-8b), reporting Geometric Volume (global) AUROC, Geometric Suspicion (local) AUARC, and ΔH over three runs.
>
> The results indicate that the method is robust to the choice of embedding model. Within every dataset/generator setting the metrics are stable across embedding models, and the 22M all-MiniLM-L6-v2 matches our 1.5B default throughout. No embedder is consistently best, and variation is driven almost entirely by task/generator model combination rather than the embedder.
>
> ## Further discussion
>
> **Inference cost.** We agree repeated sampling raises inference cost by roughly an order of magnitude. We would note this cost is inherent to *all* sampling-based UQ methods, including semantic entropy and self-consistency, rather than specific to our approach. We suggest our medical-domain results indicate the cost can be justified in high-stakes applications where errors can result in extreme consequences.
>
>
> **Uncertainty vs. hallucination.** We acknowledge that uncertainty and hallucination are not identical, and that a confidently-wrong model can evade any dispersion-based detector. Reviewer 1iGV raised a closely related concern, and the revised manuscript now treats this limitation more prominently: we close the Related Work section by stating explicitly that semantic dispersion is a proxy for hallucination risk rather than a guarantee of factual incorrectness, and we have elevated the "confidently wrong" failure mode from the appendix into the main text (Section 4.3.1 and Figure 3), showing that correction performance degrades when the sampled batch contains few correct alternatives. The Discussion further notes that white-box approaches such as factuality probes may be better suited to these cases. We welcome any further suggestions for how best to present this limitation.
>
> We hope these additions address the reviewer's concerns, and we welcome any further suggestions.

---

### Author Response · Authors · 2026-07-05
**Summary response to all reviewers**

We thank all three reviewers for their careful and constructive reviews. We are encouraged that all reviewers answered **yes** to both of TMLR's acceptance criteria — that the claims are supported by accurate and convincing evidence, and that the findings would interest TMLR's audience. We are also grateful for the reviewers' positive assessments: the geometric framework was described as a "novel perspective" (ZKc2), with the "unified treatment of global and local uncertainty," "black-box applicability," and "intuitive geometric interpretation" noted as strengths (1iGV), and the relevance of the medical and high-stakes experiments highlighted (NTGV, 1iGV).


The requested changes largely concern the *framing and presentation* of our contributions rather than their substance: tightening the language around the theory, clarifying our evaluation choices, and making the method's limitations more prominent. We have made these revisions and uploaded a revised manuscript.  The main changes are:

| Change made | Requested by |
|---|---|
| Scoped the theoretical claims to *embedding-space dispersion* under an explicit support-fidelity assumption, and added a statement clarifying that Geometric Volume is a proxy for hallucination risk, not a guarantee of factual correctness. | 1iGV |
| Added a discussion of the labelling pipeline (ROUGE-L and LLM judge), the effect of label noise on both absolute scores and method rankings, and our mitigations — supported by recent evaluation studies. | 1iGV |
| Made the threshold-tuning supervision requirement explicit in §4.2: the detector is not fully calibration-free in deployment. | 1iGV |
| Elevated the "confidently wrong" failure mode from the appendix into the main text, immediately after the local-uncertainty results. | 1iGV, NTGV |
| Added a robustness ablation across four embedding models (Appendix); the conclusions are unchanged. | NTGV |
| Clarified global vs. local uncertainty in the introduction, defining a "batch" as the responses to a *single* query. | ZKc2 |
| Stated the baseline-selection criterion explicitly in §4.1 (same black-box, sampling-based setting; spanning the main method families). | ZKc2, 1iGV |


We believe these revisions resolve the concerns raised while preserving the intended scope of the paper, and we thank the reviewers and the Action Editor for a constructive process. We explain our changes in more detail in the individual responses below, and welcome further discussion.